# On Optimal Learning Under Targeted Data Poisoning

**Steve Hanneke**
Purdue University, USA
steve.hanneke@gmail.com

**Amin Karbasi**
Yale University, USA and Google Research
amin.karbasi@yale.edu

**Mohammad Mahmoody**
University of Virginia, USA
mohammad@virginia.edu

**Idan Mehalel**
Technion, Israel
idanmehalel@gmail.com

**Shay Moran**
Technion, Israel and Google Research
shaymoran1@gmail.com

## Abstract

Consider the task of learning a hypothesis class $\mathcal{H}$ in the presence of an adversary that can replace up to an $\eta$ fraction of the examples in the training set with arbitrary adversarial examples. The adversary aims to fail the learner on a particular target test point $x$ which is *known* to the adversary but not to the learner. In this work we aim to characterize the smallest achievable error $\varepsilon = \varepsilon(\eta)$ by the learner in the presence of such an adversary in both realizable and agnostic settings. We fully achieve this in the realizable setting, proving that $\varepsilon = \Theta(\text{VC}(\mathcal{H}) \cdot \eta)$, where $\text{VC}(\mathcal{H})$ is the VC dimension of $\mathcal{H}$. Remarkably, we show that the upper bound can be attained by a deterministic learner. In the agnostic setting we reveal a more elaborate landscape: we devise a deterministic learner with a multiplicative regret guarantee of $\varepsilon \leq C \cdot \text{OPT} + O(\text{VC}(\mathcal{H}) \cdot \eta)$, where $C > 1$ is a universal numerical constant. We complement this by showing that for any deterministic learner there is an attack which worsens its error to at least $2 \cdot \text{OPT}$. This implies that a multiplicative deterioration in the regret is unavoidable in this case. Finally, the algorithms we develop for achieving the optimal rates are inherently improper. Nevertheless, we show that for a variety of natural concept classes, such as linear classifiers, it is possible to retain the dependence $\varepsilon = \Theta_{\mathcal{H}}(\eta)$ by a proper algorithm in the realizable setting. Here $\Theta_{\mathcal{H}}$ conceals a polynomial dependence on $\text{VC}(\mathcal{H})$.

## 1   Introduction

A basic goal in machine learning is to develop a predicting model from labeled examples (i.e., training data) that can reliably generalize to unseen examples (i.e., test data). In its simplest form, namely, binary classification, a learner Lrn is given a training set $S = \{(x_1, y_1), \ldots, (x_n, y_n)\}$, usually assumed to be i.i.d. samples drawn from an unknown distribution $D$ of labeled examples where $x_i$'s are the domain instances (or data points) and $y_i \in \{0, 1\}$ are the labels. The aim is to produce a mapping $h = \text{Lrn}(\mathcal{S})$ that predicts the labels of fresh examples $(x, y) \sim D$ as accurately as possible, i.e., to minimize the population loss $L_D(h) = \Pr_{(x,y)\sim D}[h(x) \neq y]$. This classical setting has been extensively studied in the last half a century. This accumulated work resulted in fundamental mathematical characterizations regarding the nature of learnability when the training samples are truly i.i.d without any tampering by an adversary [Shalev-Shwartz and Ben-David, 2014]. The goal of this paper is to offer a similar characterization in the presence of an adversary who can tamper with a subset of the training data.

36th Conference on Neural Information Processing Systems (NeurIPS 2022).

With the emergence of sensitive machine learning applications, it is critical to ensure the trustworthy of such predictive models in the non-ideal scenarios. In this paper, we consider *robust* learnability when the training examples can be altered by an adversary whose goal is to make sure that a target test point will be predicted incorrectly. For instance, a language model trained on conversations in shopping forums can be attacked by marketing campaigns, who may want a specific product to be associated with a positive experience, instead of a bad one. Another example is an adversary who aims to fool a self-driving car to speed up once it observes a stop sign. If such an adversary can somehow influence the training sets used for training the decision rules, she has all the reasons to strategically change them with the specific goal of misleading the self-driving car. As another example, consider a loan applicant who wants to make sure that his loan will be granted. If he can somehow change the training set used by the bank, he might be able to make his application approved. Note that the training set is the lens through which a learning algorithm obtains information about the underlying learning process. Therefore, once we allow the training examples to be tampered with by an adversary, even slightly, unexpected outcomes may take place. To quantify the robustness of learning algorithms, in this paper, we show how much the outcome of a learning algorithm for a particular target test can be trusted once the training set is being altered.

**(PAC) learning under instance-targeted poisoning.** More formally, we consider an adversary Adv that is allowed to *replace* an $\eta$-fraction of the training sample $S$, resulting to a tampered training sample $S'$ given to the learning algorithm Lrn. Note that even though the training sample $S$ is drawn i.i.d. from a distribution $D$, the tampered training sample $S'$ does not enjoy this property anymore. Such attackers are also called *poisoning* adversaries [Barreno, Nelson, Sears, Joseph, and Tygar, 2006], and variants of them are previously studied under the name of *malicious* noise [Valiant, 1985, Kearns and Li, 1993] or *nasty* noise [Bshouty, Eiron, and Kushilevitz, 2002]. More specifically, we study poisoning settings in which the adversarial perturbation of the original sample $S$ *can also depend on the final test instance $x$*. Due to the adversary's knowledge of the target test point $x$, such poisoning attacks are sometimes referred to as *instance-targeted* poisoning attacks [Barreno, Nelson, Sears, Joseph, and Tygar, 2006]. Even without any manipulation to the training set, it is too much to ask the learning algorithm to predict correctly all the time while given only a finite number of examples to learn from. In the same vein, we can only hope to design a robust learning algorithm that is correct with high probability over the selection of $(x, y) \sim D$, especially if the adversary knows the test instance $(x, y)$ before manipulating the training set $S$ to $S'$. Gao, Karbasi, and Mahmoody [2021], building on ideas from [Levine and Feizi, 2020], proved that PAC learnability under instance-targeted poisoning attacks is achievable only when $\eta = o(1)$. In other words, when the adversary can only change a *sublinear* $o(n)$ number of $n$ examples, then the optimal learner can achieve error $o(1)$ that goes to zero when the number of examples $n$ goes to infinity.

## 1.1 Our Results

The prior work leaves several key questions open on the exact parameters of learnability under instance-targeted poisoning. Most importantly, the work of Gao, Karbasi, and Mahmoody [2021] does not quantify the error rate when the adversary's budget is $\eta = \Omega(1)$ (e.g., if the adversary can corrupt $n/100$ of the examples). Secondly, Gao, Karbasi, and Mahmoody [2021] only assume the realizable setting as it is crucial for their results that all the "sub-models" trained using the bagging technique will have error that goes to *zero*. Hence, the question of finding optimal learning rates is left open for both realizable and agnostic settings. Finally, as the developed robust algorithms are all based on "bagging" they are inherently improper learning technique.

In this work, we make progress on all the directions above and achieve optimal error rates (up to constant factors) for general $\eta$, both for the realizable and agnostic settings. We further study the proper nature of the obtained algorithms and give the first proper learning methods that are robust against instance-targeted poisoning attacks for natural hypothesis classes such as linear classifiers. More precisely, we give a characterization of the optimal error rate of learning under instance-targeted poisoning attacks with budget $\eta \cdot n$ as follows.

**Realizable setting.** We show that the optimal error is $\Theta(\eta \cdot d)$ where $d$ is the VC dimension of the hypothesis set $\mathcal{H}$. To prove this, we first present an upper bound, showing that a (deterministic) learner can guarantee the error to be at most $O(\eta d)$ under any instance-targeted poisoning attacks of budget $\eta n$. We then also show a matching lower bound (up to a constant factor) as follows. For any

*nontrivial*[1] hypothesis class of VC dimension $d$, we show how to design a distribution $D$ over the examples such that no matter how the learning proceeds, there always exists an adversary of budget $\eta n$ that can increase the error (under the instance-targeted attack) to $\Omega(\eta d)$. Our lower bound above holds even if the learning algorithm uses *private randomness* that is not known to the adversary[2]. Our positive result, however, is deterministic, and so can be seen as satisfying the *stronger* guarantee, in which the adversary's perturbations to the training set is allowed to depend on learner's randomness.

**Agnostic setting.** We also extend our result above to the agnostic setting in which all hypotheses $h \in \mathcal{H}$ have population loss bounded away from zero (even before the attack). In this setting, we devise a deterministic algorithm whose expected error on the test point is $O(\mathsf{OPT} + \eta \cdot d)$, where $\mathsf{OPT}$ is the population loss of the best hypothesis $h \in \mathcal{H}$.

A natural question that arises is whether one can achieve an additive regret guarantee of $\mathsf{OPT} + O(\eta \cdot d)$? (Note that agnostic learning is usually defined with respect to additive regret). We show that this is in fact *not possible*, at least for deterministic learners, by presenting a negative result. In particular we show that for any deterministic learner Lrn, there is an extremely simple hypothesis class (just consisting of two functions) and an input distribution such that the learner is forced to have adversarial error $\geq 2\mathsf{OPT}$. This negative result uses tools from the computational concentration of products [Talagrand, 1995] and a continuity intermediate-value argument.

**Proper learning.** The deterministic algorithm witnessing the above upper bound is inherently improper which might be a disadvantage in terms of interpretability or test-time computational complexity. In contrast, in (the non-adversarial) PAC setting proper algorithms are known to achieve near optimal learning rates (up to log factors). We therefore explore the cost of proper learning under instance-targeted poisoning attacks. We show that in many natural classes, such as half spaces, it is indeed possible to obtain proper learning rules that are robust to instance-targeted poisoning attacks, with guarantees which are only polynomially worse than optimal. For example, for the class of half-spaces in $\mathbb{R}^d$ we derive a deterministic proper learning rule whose error rate is at most $O(d^3 \eta)$. At a technical level, we achieve this result by relying on the *projection number* of the class [Bousquet, Hanneke, Moran, and Zhivotovskiy, 2020, Kane, Livni, Moran, and Yehudayoff, 2019, Braverman, Kol, Moran, and Saxena, 2019].

## 1.2 Relation to Certification and Stability

**Certification.** Robustness to instance-targeted poisoning boils down to the following type of stability: on most of the test instances $x$, the prediction of the learner $y = y(x)$ remains the same even if at most $\eta$ fraction of the examples in the training-set $S$ are replaced. It is natural to require the learning rule to *certify* this stability. That is, a certifying learning rule provides a bound $k = k(x)$ along with the prediction label $y = y(x)$, where the meaning of $k$ is that the prediction $y = y(x)$ remains the same even if at most $k$ examples in the input sample are replaced. Note that it is always possible to provide the trivial guarantee of $k = 0$, and therefore the goal is to design robust learners that provide non-trivial certificates. Our algorithm naturally achieves that: for $\approx 1 - \varepsilon$ of the test instances $x$ it provides a guarantee of $k \approx \eta n$.

**Connection to stability.** We also present a new perspective on instance-targeted poisoning attacks by showing how they can be seen as natural forms of algorithmic stability [Bousquet and Elisseeff, 2002, Rakhlin, Mukherjee, and Poggio, 2005]. In particular, we show that one can study the adversarial robustness (around the *true* label) to instance-targeted poisoning by decoupling the (pure) stability aspect (which does not depend on the true labels) from the (non-adversarial) risk. We refer to the former as the *prediction stability*. Roughly speaking, prediction stability requires that the model's prediction on $x$ does not change even if the adversary changes the training set withing its budget $\eta n$. Note that here we do not care whether the model's output on $x$ is the correct label or not, and hence is a pure measure of stability of the predictions.

It might be helpful to compare prediction stability with the algorithmic stability of [Bousquet and Elisseeff, 2002, Rakhlin, Mukherjee, and Poggio, 2005]. The later requires that for a typical sample $S$ of size $n$, and for every *fixed* $i \in [n]$, the prediction of the model trained on $S$ and tested on a random

---

[1]A non-trivial class $\mathcal{H}$ is one for which there are $x_1, x_2 \in \mathcal{X}$ and $h_1, h_2 \in \mathcal{H}$ so that $h_1(x_1) = h_2(x_1)$ and $h_1(x_2) \neq h_2(x_2)$. In particular, any class containing at least 3 hypotheses is non-trivial.

[2]This model is referred to as the "weak" learning model (under instance-targeted poisoning attacks) in the work of Gao, Karbasi, and Mahmoody [2021].

test-point $x$ is likely not changed if one substitutes the $i$-th example in $S$ with a *fresh* random example. Prediction stability strengthens this condition in two ways: (1) the choice of what coordinate in $S$ to change can adversarially depend on the test instance $x$, (2) the adversary is allowed to change *more* than one examples (i.e., up to $\eta \cdot n$).

## 1.3 Related Work

Poisoning attacks are studied in theoretical learning under various noise models [Valiant, 1985, Kearns and Li, 1993, Sloan, 1995, Bshouty, Eiron, and Kushilevitz, 2002]. However, these works focus on the *non-targeted* setting in which the adversary does *not* know the target instance.

The *computational* aspects of efficient learning under (non-targeted) poisoning have been studied in various works, including that of Kalai, Klivans, Mansour, and Servedio [2008], Klivans, Long, and Servedio [2009], Awasthi, Balcan, and Long [2014], with this last work obtaining nearly optimal (up to constants) learning guarantees among polynomial-time algorithms for learning homogeneous linear separators with malicious noise under distribution restrictions. That result was subsequently extended to the *nasty noise* model by Diakonikolas, Kane, and Stewart [2018], via techniques that also enable them to study other geometric concept classes. In the *unsupervised* setting, Diakonikolas, Kamath, Kane, Li, Moitra, and Stewart [2016], Lai, Rao, and Vempala [2016] studied the computational aspect of learning under poisoning. In contrast, our work focuses on (supervised) instance-targeted poisoning, and we study the learning rates *information theoretically* regardless of learner's computing power. The work of Steinhardt, Koh, and Liang [2017] further studied the certification of the overall (non-targeted) error. More recently, such (non-targeted) poisoning attacks are combined with *test-time* attacks and are studied under the name of *backdoor* attacks [Gu, Dolan-Gavitt, and Garg, 2017, Ji, Zhang, and Wang, 2017].

Besides instance-targeted attacks (which are the focus of this paper), other notions of targeted attacks were studied in the literature: for example, in *model-targeted* attacks, the adversary's goal is to make the learner predict according to a specific model. Recent works on this model include [Farhadkhani, Guerraoui, Hoang, and Villemaud, 2022, Suya, Mahloujifar, Suri, Evans, and Tian, 2021]. Some other works study *label-targeted* attacks, in which the adversary's goal is to flip the decision on the test instance to a specific label (e.g., see *targeted misclassification* attacks in [Chakraborty, Alam, Dey, Chattopadhyay, and Mukhopadhyay, 2018]). The work of [Jagielski, Severi, Pousette Harger, and Oprea, 2021] studies a generalization of instance-targeted attacks, called *subpopulation* attacks, in which the adversary knows the subset of the inputs, from which the test instance will be drawn.

Most relevant to our setting are the recent works of Gao, Karbasi, and Mahmoody [2021], Blum, Hanneke, Qian, and Shao [2021] where the general problem of learning (and more quantitative variant of learning error rate) under *instance-targeted* poisoning was formally defined and studied. In particular, Blum, Hanneke, Qian, and Shao [2021] studied learnability under instance-targeted poisoning where the adversary can add an *unbounded* number of so-called clean-label examples to the training set. A clean-label example $(x, y)$ has the property that $y$ is the *correct* label of $x$, while $x$ could be an arbitrary instance that is *not* sampled from the same distribution that generates other instances in the training set. Gao, Karbasi, and Mahmoody [2021] also showed that when the adversary's corruption is only an $o(1)$ fraction of the training set, PAC learning is possible (if it is possible without the attack). In a concurrent work, Balcan, Blum, Hanneke, and Sharma [2022] study the problem of certifying the *correct* prediction even under instance-targeted data poisoning. Our methods, however, can be used to obtain certification of the stability of the model around their prediction (even though the prediction might *not* be true always), while controlling the overall error to be provably small (again under the instance-targeted attack).

Rosenfeld, Winston, Ravikumar, and Kolter [2020] empirically demonstrated that randomized smoothing [Cohen, Rosenfeld, and Kolter, 2019] can provide robustness against label-flipping attacks, in which the adversary is limited to merely flipping the label of a subset of the training set. They also showed that randomized smoothing can be used to handle *replacing* attacks (the model also studied in this paper), in which the adversary substitutes a part of the training set with a new set of same size. Subsequently, Levine and Feizi [2020] used deterministic methods that further allowed attacks that can add examples to or remove them from the training set. Chen, Li, Wu, Sheng, and Li [2020], Weber, Xu, Karlas, Zhang, and Li [2020], Jia, Cao, and Gong [2020] further developed the technique of randomized bagging/sub-sampling for the goal of resisting instance-targeted poisoning attacks.

Finally, we comment that other theoretical works have also studied instance-targeted poisoning attacks [Mahloujifar and Mahmoody, 2017, Etesami, Mahloujifar, and Mahmoody, 2020]. These works show how to *amplify* error for specific test instances, say from 0.01 error to 0.5, through instance-targeted poisoning. In particular, these works do not talk about the *fraction* of the test population that is vulnerable to targeted poisoning. The work of Shafahi, Huang, Najibi, Suciu, Studer, Dumitras, and Goldstein [2018] studied the power of such attacks empirically.

## 2 Preliminaries

**Notation and basic learning theory definitions.** We consider the setting of binary classification. Let $\mathcal{X}$ denote the input domain and $\mathcal{Y} = \{0, 1\}$ denote the label-set. A pair $(x, y) \in \mathcal{X} \times \mathcal{Y}$ is called an *example*. A sequence $S = (x_1, y_1), \ldots, (x_n, y_n) \in (\mathcal{X} \times \mathcal{Y})^n$ of $n$ examples is a *sample* of size $n$. The $i$'th example in $S$ is denoted by $S_i$.

A function $h \colon \mathcal{X} \to \mathcal{Y}$ is called an hypothesis or a concept. A set of hypotheses $\mathcal{H} \subset \mathcal{Y}^{\mathcal{X}}$ is called an *hypothesis class*, or a *concept class*. We denote the VC-dimension of a concept class $\mathcal{H}$ by $d = d(\mathcal{H})$.

For a set $Z$, let $Z^* = \cup_n Z^n$ denote the set of all finite sequences with elements from $Z$. A *learning rule* or *learning algorithm* or *learner* $\mathsf{Lrn} \colon (\mathcal{X} \times \mathcal{Y})^* \to \mathcal{X}^{\mathcal{Y}}$ is a deterministic[3] mapping which takes an input sample $S \in (\mathcal{X} \times \mathcal{Y})^*$ and maps it to a hypothesis $\mathsf{Lrn}(S) = h \in \mathcal{X}^{\mathcal{Y}}$. If it is guaranteed that $\mathsf{Lrn}(S) \in \mathcal{H}$ for all input samples $S$ then $\mathsf{Lrn}$ is said to be *proper*; otherwise, it is *improper*.

Let $D$ be a distribution over examples, and let $h$ be an hypothesis. The *population loss* of $h$ with respect to $D$ is defined by $L_D(h) = \Pr_{(x,y) \sim D}[h(x) \neq y] = \mathbb{E}_{(x,y) \sim D}[1[h(x) \neq y]]$. A distribution $D$ is said to be *realizable* by $\mathcal{H}$ if $\inf_{h \in \mathcal{H}} L_D(h) = 0$. Similarly, for a sample $S$, let $L_S(h) = \frac{1}{|S|} \sum_{i=1}^n 1[h(x_i) \neq y_i]$ denote the *empirical error* of $h$ with respect to $S$, and call a sample realizable by a class $\mathcal{H}$ if there exists $h \in \mathcal{H}$ such that $L_S(h) = 0$. The expected loss (also called risk) of a learning algorithm $\mathsf{Lrn}$ w.r.t a distribution $D$ and sample size $n$ is defined by $\varepsilon_n(\mathsf{Lrn}|D) := \Pr_{S \sim D^n, (x,y) \sim D}[\mathsf{Lrn}(S)(x) \neq y]$. The function $n \mapsto \varepsilon_n(\mathsf{Lrn}|D)$ is called the *learning curve*, or *learning rate* of $\mathsf{Lrn}$ w.r.t $D$.

For a real number $r$, let $\lfloor r \rceil$ denote the nearest integer to $r$. In case of ties, when $r = k + 1/2$ for some $k \in \mathbb{Z}$, then define $\lfloor r \rceil = k + 1$. For any finite multiset $\mathcal{H}' \subset \mathcal{H}$, denote by $\mathsf{Maj}(\mathcal{H}')$ the function defined for all $x \in \mathcal{X}$ by $\mathsf{Maj}(\mathcal{H}')(x) = \left\lfloor \frac{1}{|\mathcal{H}'|} \sum_{h' \in \mathcal{H}'} h'(x) \right\rceil$.

**Adversarial risk and prediction stability.** Before we introduce the definition of Adversarial risk, we define *Hamming distance* between samples, which is a natural way to quantify distance between samples of equal size.

**Definition 2.1** (Hamming distance between samples). Fix $n \in \mathbb{N}$ and let $S, S' \in (\mathcal{X} \times \mathcal{Y})^n$. We define the *Hamming distance* between $S$ and $S'$ by $\mathsf{d}_{\mathsf{H}}(S, S') = \sum_{i=1}^n 1[S_i \neq S'_i]$.

Note that the Hamming distance is defined only for samples of equal sizes. If $\mathsf{d}_{\mathsf{H}}(S, S') \leq \eta \cdot n$, we say that $S, S'$ are $\eta$-*close*. For any sample $S$, let $B_\eta(S) := \{S' : \mathsf{d}_{\mathsf{H}}(S, S') \leq \eta \cdot n\}$.

**Definition 2.2** ($\eta$-adversarial risk). Let $\eta \in (0, 1)$ be the adversary's budget, let $\mathsf{Lrn}$ be a learning rule, and let $D$ be a distribution over examples. The $\eta$-*adversarial risk* of $\mathsf{Lrn}$ w.r.t $D$ and sample size $n$ is defined by

$$\varepsilon_n^{\mathsf{Adv}}(\mathsf{Lrn}|D, \eta) := \Pr_{S \sim D^n, (x,y) \sim D} \left[ \exists S' \in B_\eta(S) : \mathsf{Lrn}(S')(x) \neq y \right].$$

Thus, robust learning with respect to instance-targeted poisoning with budget $\eta$ boils down to minimizing the adversarial risk. Indeed, given an input sample $S$ and a test example $(x, y)$, an adversary with budget $\gamma$ can force a mistake on $x$ if and only if $\mathsf{Lrn}(S')(x) \neq y$ for some $S' \in B_\eta(S)$.

**Randomness.** In Definition 2.2 above we define adversarial risk for the setting in which both the learner $\mathsf{Lrn}$ and the model $h = \mathsf{Lrn}(S)$ are *deterministic*. When either $\mathsf{Lrn}$ or $h$ is allowed to use randomness, then the notion of adversarial risk as defined in Definition 2.2 can be extended in several ways, depending on whether the adversary can see the randomness of the learner or not. Some of these variations are discussed in the work of Gao, Karbasi, and Mahmoody [2021]. We remark

---

[3]In Appendix B, we extend the definition in a way that captures also a family of randomized learners.

however that our results in the realizable setting apply to all variations. This is simply because our upper bounds are achieved by deterministic learners, whereas our lower bound uses the weakest type of an adversary (which does not depend on the randomness of the learner). In contrast, our lower bound in the agnostic setting applies only to deterministic learners.

**Explicit bounds.** We do not try to optimize the constants hidden in the $O(\cdot), \Omega(\cdot)$ notation in the derived bounds. The reason is because on the one hand, this way the proofs are simpler and more accessible, and on the other hand, we do not know how to get tight (or nearly tight) lower and upper bounds on the constants. Obtaining tight bounds is a natural direction for future research; we elaborate on this in Section 5. Nevertheless, the complete proofs (which are given in the appendix) include explicit numerical bounds on the constants.

**Decoupling adversarial risk into stability and risk.** It is convenient and illustrative to decouple robust learnability to two properties: small expected loss and prediction stability. The latter means that the prediction of the learning algorithm on a random test point is stable under replacing a bounded amount of examples from the training set:

**Definition 2.3** (Prediction stability). Let $n \in \mathbb{N}$, $\sigma, \eta \in (0,1)$. Let Lrn be a learning rule and $D$ be a distribution over examples. We say that the learning rule Lrn is $(n, \sigma, \eta)$-*prediction stable with respect to $D$* if the following holds

$$\lambda_n(\mathsf{Lrn}|D,\eta) := \Pr_{S \sim D^n, x \sim D_x}[\exists S' \in B_\eta(S) : \mathsf{Lrn}(S')(x) \neq \mathsf{Lrn}(S)(x)] \leq \sigma.$$

where $D_x$ is the marginal distribution induced by $D$ on the domain $\mathcal{X}$.

Of course, prediction stability alone does not guarantee robust learning. Indeed, useless learning rule that always outputs the all 0's classifier has maximal stability. At the very least, the learning rule should learn the class in the classical sense (in the absence of an adversary). The following observation asserts that prediction-stable learning rules with small loss are robust learners:

**Observation 2.4** (Prediction stability + small error = robust learning). *Let* Lrn *be a learner and $D$ a distribution over examples. Then,*

$$\max\{\varepsilon_n(\mathsf{Lrn}|D), \lambda_n(\mathsf{Lrn}|D,\eta)\} \leq \varepsilon_n(\mathsf{Lrn}|D,\eta) \leq \lambda_n(\mathsf{Lrn}|D,\eta) + \varepsilon_n(\mathsf{Lrn}|D).$$

In other words, if Lrn is $(n, \sigma, \eta)$-prediction stable with respect to $D$ whose expected population loss is $\varepsilon_n(\mathsf{Lrn}|D) \leq \varepsilon$. Then Lrn learns $D$ with an adversarial expected loss $\sigma + \epsilon$. Conversely, if $\varepsilon_n(\mathsf{Lrn}|D,\eta) \leq \varepsilon$ then Lrn is $(n, \varepsilon, \eta)$-prediction stable with respect to $D$ and its expected population loss is also $\varepsilon_n(\mathsf{Lrn}|D) \leq \varepsilon$. We leave the (simple) proof of Observation 2.4 to the reader.

## 3 Realizable Setting

Theorems 3.1 and 3.3 below characterize the optimal adversarial risk in the realizable setting.

**Theorem 3.1** (Realizable case – positive result). *There exists a constant $c_1 > 0$ so that the following holds. Let $\mathcal{H}$ be a hypothesis class with VC dimension $d$ and let $\eta \in (0,1)$. Then there exists a learner* Lrn *having $\eta$-adversarial risk*

$$\varepsilon_n^{\mathsf{Adv}}(\mathsf{Lrn}|D,\eta) \leq c_1\eta d$$

*for any distribution $D$ realizable by $\mathcal{H}$ and for any sample size $n \geq 1/\eta$.*

We prove Theorem 3.1 in Appendix A.

Note that the requirement that the sample size is $n \geq 1/\eta$ is necessary since otherwise $\eta \cdot n < 1$, which means that the adversary cannot modify the input sample, and so this case reduces to classical learning without an adversary.

Theorem 3.1 is proven using the STABLE PARTITION AND VOTE (or SPV, for short) meta-algorithm, described in Figure 1. The meta-algorithm is based on the idea of partitioning and then voting used in [Gao, Karbasi, and Mahmoody, 2021], but with a more refined and precise analysis. The partition and vote technique works as follows. First, partition the input sample to subsamples of a carefully chosen size. Then, train a given learner (which is called the *input learner* of SPV) on each subsample, and finally let the trained learners vote to determine the output label. The size of each subsample

SPV: Stable Partition and Vote

**Input:** Stability parameter $\eta \in (0,1)$, a learning algorithm Lrn and an input sample $S \sim D^n$ where $n \geq 1/\eta$.
**Output:** A classifier $h \colon \mathcal{X} \to \mathcal{Y}$.

1. Partition $S$ into $\lceil 7\eta n \rceil$ consecutive subsamples such that all first $t = \lfloor 7\eta n \rfloor$ subsamples are of size at least $\frac{1}{7\eta}$. Denote the $i$'th subsample by $S^{(i)}$.

2. For all $i \in [t]$, run the learning algorithm Lrn on $S^{(i)}$ to obtain a hypothesis $h_i = \mathsf{Lrn}(S^{(i)})$.

3. Return the hypothesis $h$ defined as follows for all $x \in \mathcal{X}$ :
$$h(x) = \mathsf{Maj}\left(\{h_1, \ldots, h_t\}\right)(x).$$

Figure 1: SPV - A meta algorithm implementing a stable version of the input learning algorithm Lrn.

trades-offs, in a way, expected loss and prediction-stability: if it is too small, the given learner will perform poorly on each subsample. On the other hand, if it is relatively large then the number of learners that participate in the majority vote is small and the adversary can poison a large fraction of these learners and flip the overall majority vote. We elaborate on this when proving Theorem 3.1. Notice that the time complexity of SPV is proportional to the time complexity of the learner Lrn.

To state the complementing impossibility result, we need the following definition of *non-trivial concept classes* [Bshouty, Eiron, and Kushilevitz, 2002].

**Definition 3.2** (Non-trivial concept classes)**.** We say that a concept class $\mathcal{H}$ over a domain $\mathcal{X}$ is *non-trivial*, if there are $x_1, x_2 \in \mathcal{X}$ and $h_1, h_2 \in \mathcal{H}$ so that $h_1(x_1) = h_2(x_1)$ and $h_1(x_2) \neq h_2(x_2)$.

**Theorem 3.3** (Realizable case – impossibility result)**.** *There exists a constant $c_2 > 0$ so that the following holds. Let $\mathcal{H}$ be a non-trivial hypothesis class with VC dimension $d$ and let $\eta \in (0,1)$. Then, there exists a distribution $D$ realizable by $\mathcal{H}$, so that every learner Lrn has $\eta$-adversarial risk*

$$\varepsilon_n^{\mathsf{Adv}}(\mathsf{Lrn}|D, \eta) \geq \min\{c_2 \eta d, 1/100\}$$

*for any sample size $n \geq 1/\eta$.*

We note that this impossibility result applies also to a variety of randomized learners; we elaborate on this in Appendix B, where we also prove Theorem 3.3.

The above lower bound demonstrates how vulnerability to instance-targeted attacks depends greatly on the hypothesis class we want to learn, and specifically on its VC-dimension.

## 3.1 Certification

Besides prediction-stability, another useful property our SPV meta-algorithm has is the ability to efficiently calculate and output a *certificate* for the stability of its predictions. Formally, given an input sample $S$, a certificate is a function $\eta_S : \mathcal{X} \to [0,1]$, outputted by a learner in addition to its output hypothesis $h_S$ such that the following is satisfied: $h_S(x) = h_{S'}(x)$ for every point $x$ and for every input sample $S'$ which is $\eta_S(x)$-close to $S$. If one ignores computational considerations, outputting optimal certificates is always possible:

**Definition 3.4** (Optimal Certificate)**.** Let Lrn be any learning rule, and let $S$ be an input sample. Define the *optimal certificate* $\eta^\star(\cdot) = \eta^\star(\cdot|S)$ of Lrn on input sample $S$ as follows. The optimal certificate $\eta^\star(x|S)$ is equal to $\frac{k}{n}$ where $k$ is the largest integer for which $\mathsf{Lrn}(S')(x) = \mathsf{Lrn}(S)(x)$ for every sample $S'$ with hamming distance at most $k$ from $S$.

In other words, if $S$ is a sample that was corrupted by an adversary with budget $\eta$ such that $\eta \leq \eta^\star(x|S)$ then the output label $\mathsf{Lrn}(S)(x)$ is equal to the label that would have been outputted if the learner was trained with the uncorrupted sample.

The issue with the optimal certificate $\eta^\star(x)$ is that it can be impossible to compute as it requires to iterate over the potentially infinite space of all samples $S'$ of hamming distance at most $n \cdot \eta(x)$ from



PSPV: PROPER STABLE PARTITION AND VOTE

**Input:** Stability parameter $\eta \in (0, 1)$, a proper learning algorithm $\mathsf{Lrn}_p$ and an input sample $S \sim D^n$ where $n \geq 1/\eta$.

**Output:** A classifier $h \in \mathcal{H}$.

1. Partition $S$ into $\lceil 5k_p\eta n \rceil$ consecutive subsamples such that all first $t = \lfloor 5k_p\eta n \rfloor$ subsamples are of size at least $\frac{1}{5k_p\eta}$. Denote the $i$'th subsample by $S^{(i)}$.

2. For all $i \in [t]$, train $\mathsf{Lrn}_p$ on $S^{(i)}$ to obtain a hypothesis $h_i = \mathsf{Lrn}_p(S^{(i)})$.

3. Return $h \in \mathcal{H}$ such that

$$h(x) = \mathsf{Maj}\left(\{h_1, \ldots, h_t\}\right)(x)$$

holds for all $x \in \mathcal{X}_{\{h_1,\ldots,h_t\},2k_p}$.



Figure 2: PSPV - A meta-algorithm that implements a stable version of the input proper learning algorithm $\mathsf{Lrn}_p$ and maintains properness.

the input sample $S$. In contrast, our SPV learner can *efficiently* calculate a non-trivial lower bound on $\eta^\star$ which therefore also serves as a certificate. The key property which enables this is the fact that its output hypothesis is the majority vote of base learners, each trained on a *disjoint* subsample. This is summarized in the following proposition:

**Proposition 3.5.** *Consider a learner whose output hypothesis is given by a majority vote of $t$ learners $L_1, \ldots, L_t$ that are trained on $t$ disjoint subsamples $S_1, \ldots, S_t$ of the input sample $S$. Define*

$$\eta(x|S) = \frac{1}{n} \cdot \left( \frac{\sum_{i\in[t]} 1[h_i(x) = y] - \sum_{i\in[t]} 1[y_i \neq y]}{2} - 1 \right),$$

*where $h_i$ is the output hypothesis of $L_i$, $y$ is the output label of the majority vote of the $L_i$'s, and $n$ is the size of the input sample $S$. Then, $\eta(x|S) \leq \eta^\star(x|S)$.*

*Proof.* Notice that $n \cdot \eta(x|S) + 1$ is equal to the minimal number of $h_i$'s whose prediction on $x$ must be flipped in order to enforce that $|\{i : h_i(x) = y\}| \leq |\{i : h_i(x) \neq y\}|$. Therefore, at least one example in each $S_i$ such that $h_i(x) = y$ must be replaced in order to change the prediction of $\mathsf{Lrn}(S)$ on $x$. In particular, if only $n \cdot \eta(x|S)$ examples are replaced than the prediction of $\mathsf{Lrn}(S)$ on $x$ remains the same. This implies that $\eta^\star(x|S) \geq \eta(x|S)$ as stated. $\qquad\square$

In light of Proposition 3.5, our SPV learner can efficiently compute and output a certificate $\eta(x)$ which is proportional to $\eta$ (where $\eta$ is the stability parameter given to SPV), with probability proportional to the expected loss of the input learner given to SPV when executed on a sample of size $\left\lceil \frac{1}{7\eta} \right\rceil$.

### 3.2 A Proper Variant of SPV

We now present a proper version of SPV for classes $\mathcal{H}$ with a finite *projection number*, described in Figure 2. The projection number of a concept class $\mathcal{H}$ is denoted by $k_p = k_p(\mathcal{H})$ (we present its definition after the statement of Theorem 3.6 below). In particular, for the class of halfspaces it yields a robust learner with the following guarantee:

**Theorem 3.6.** *There exists a constant $c > 0$ so that the following holds. Let $\mathcal{H}$ be the class of halfspaces over $\mathbb{R}^d$ for some $d \geq 1$, and let $\eta \in (0, 1)$. Then, there exists a proper learner $\mathsf{Lrn}$ having $\eta$-adversarial risk*

$$\varepsilon_n(\mathsf{Lrn}|D, \eta) \leq c\eta d^3$$

*for any distribution $D$ realizable by $\mathcal{H}$ and for any sample size $n \geq 1/\eta$.*

The proof of Theorem 3.6 is deferred to Appendix C.

To derive Theorem 3.6, we reinforce the SPV algorithm with a technique introduced by Kane, Livni, Moran, and Yehudayoff [2019] and further developed by Bousquet, Hanneke, Moran, and

Zhivotovskiy [2020]. This technique allows in certain cases to *project* a majority vote of hypotheses from the class $\mathcal{H}$ back to $\mathcal{H}$. Its applicability hinges on a combinatorial parameter called the *projection number*. The PSPV learner explicitly uses the projection number, so for completeness we give its definition below. The interested may see the work of Bousquet, Hanneke, Moran, and Zhivotovskiy [2020] for an insightful discussion on the role of the projection number in proper learning.

**Definition 3.7** (Projection Number). Let $\mathcal{H}$ be a concept class. For any $\ell \geq 2$ and for any multiset $\mathcal{H}' \subset \mathcal{H}$ define the set $\mathcal{X}_{\mathcal{H}',\ell}$ to be the set of all $x \in \mathcal{X}$, for which the number of hypotheses in $\mathcal{H}'$ that disagree with $\mathsf{Maj}(\mathcal{H}')(x)$ is less than $|\mathcal{H}'|/\ell$. The Projection Number of the class $\mathcal{H}$, denoted $k_p = k_p(\mathcal{H})$, is defined to be the smallest $\ell$ so that for any finite multiset $\mathcal{H}' \subset \mathcal{H}$, there exist $h \in \mathcal{H}$ such that $h(x) = \mathsf{Maj}(\mathcal{H}')(x)$ for all $x \in \mathcal{X}_{\mathcal{H}',\ell}$. If no such $\ell$ exists then $k_p = \infty$.

# 4 Agnostic Setting

In this section, we extend the results on robust learnability to the agnostic case. First, by a simple generalization of the positive result for the realizable case, we provide a robust semi-agnostic learner. That is, our learner has adversarial risk depending linearly on $\mathsf{OPT} = \mathsf{OPT}(\mathcal{H}, D) := \min_{h \in \mathcal{H}} L_D(h)$. While semi-agnostic learning is considered not ideal in many cases, we complement our positive result by showing that semi-agnostic learning is unavoidable when the goal is to design a robust and deterministic (as ours) learner for the agnostic setting.

## 4.1 A Semi-agnostic Learner

Formally, a semi agnostic learner is defined as follows. Let $c \in \mathbb{R}$. A learning rule Lrn is a *c-semi agnostic learner* if the following holds. Let $\mathcal{H}$ be a concept class and let $D$ be a distribution over examples. Then there exists an *excess error rate* $\varepsilon^{\mathsf{Agn}} : \mathbb{N} \to [0, 1]$ such that $\varepsilon_n(\mathsf{Lrn}|D) \leq c\mathsf{OPT} + \varepsilon^{\mathsf{Agn}}(n)$ where $\mathsf{OPT} = \inf_{h \in \mathcal{H}} L_D(h)$.

Before stating our positive result in this setting, we first discuss how achieving adversarial risk $O(d(\mathsf{OPT} + \eta))$ is possible by reduction to the realizable setting.

**Reduction to the realizable setting.** Suppose a learner is given a training set $S'$ of size $n$ that comes with $\eta n$ replacements made by the adversary on the original set $S$. Moreover, suppose that $S$ is sampled from a distribution $D$ such that the best $h \in \mathcal{H}$ has $\mathsf{OPT}$ error on $D$. This means that, roughly $\mathsf{OPT}$ fraction of $S$ does *not* match to $h$. Therefore, one can see $S'$ as first sampled from $D$ (without noise) followed by $\approx (\eta + \mathsf{OPT}) \cdot n$ replacement corruptions. This way, one can employ a learner that can tolerate $\eta' = \eta + \mathsf{OPT}$ fraction of adversarial corruptions in the realizable setting and obtain total adversarial risk $O(d(\mathsf{OPT} + \eta))$.

The above discussion raises a natural question: can a learner achieve adversarial risk $O(\mathsf{OPT} + d\eta)$ or even (ideally) $\mathsf{OPT} + O(d\eta)$? The latter is the typical type of risk bound in agnostic settings, where there is no multiplicative dependence on $\mathsf{OPT}$ in the risk.

The following theorem, which we prove in Appendix D states the positive result.

**Theorem 4.1** (Positive result for the agnostic case). *There exist constants $c_1, c_2$ so that the following holds. Let $\mathcal{H}$ be a hypothesis class with VC dimension $d$ and let $\eta \in (0, 1)$. Then, there exists a learner Lrn having $\eta$-adversarial risk*

$$\varepsilon_n^{\mathsf{Adv}}(\mathsf{Lrn}|D, \eta) \leq c_2 \cdot \mathsf{OPT} + c_1 \cdot d \cdot \eta$$

*for any distribution $D$ over examples and for any sample size $n \geq 1/\eta$.*

As in the realizable case upper bound, the above upper bound is proved by using the SPV meta-learner. The main difference is that to prove this result we use a different input learner Lrn given to SPV than the one we use in the realizable case.

## 4.2 Ruling Out Agnostic Learning

Note that Theorem 4.1 only proves the existence of a *semi*-agnostic learner under instance-targeted poisoning. A more desirable goal would be to obtain (standard) *agnostic* learners whose error under the attack is $\mathsf{OPT} + \psi$ where $\psi$ is a vanishing (additive) error term when $\eta \to 0$. Here we will prove that at least when it comes to *deterministic* learners, such a goal is out of reach, and the best we can

hope for is 2OPT plus additive terms that depend on $\eta$ and the VC dimension. This explains why we can only achieve a semi-agnostic learner.

The following theorem, which we prove in Appendix E shows that in Theorem 4.1, the constant $c_1$ needs to be at least 2, and so the standard way agnostic learners bound their regret is not possible for instance-targeted poisoning.

**Theorem 4.2** (Impossibility of agnostic learning). *Let $\eta' \in (0,1), n \in \mathbb{N}$. For any hypothesis class $\mathcal{H}$ that has at least two hypotheses and for any deterministic learner, there is a distribution $D$ over (two) examples and $\eta = \eta' + \widetilde{O}(1/\sqrt{n})$ such that Lrn has $\eta$-adversarial risk*

$$\varepsilon_n^{\mathsf{Adv}}(\mathsf{Lrn}|D,\eta) \geq 2\mathsf{OPT} + \Omega(\eta') - O(1/n).$$

## 5 Conclusion and Open Questions

In this work, we studied the optimal rate of learning for binary classification problems under instance-targeted poisoning. We showed that in the realizable setting the error rate can be characterized up to a constant factor and is proportional both to adversary's budget and the VC dimension of the class. In the agnostic setting, we proved a perhaps surprising lower bound that standard agnostic learning (with additive regret compared to the optimal error in the no-attack setting) is impossible for deterministic learners, and also complemented this with a positive result using a *semi-agnostic* learner. We also showed how to make our learners proper in a variety of interesting settings.

Our work leaves a few interesting directions for future research.

- **Finding the exact constant in the realizable case.** Our results in the realizable case characterize the optimal adversarial risk up to a constant multiplicative factor in the sense that there exist constants $c_1, c_2$ so that achieving $\eta$-adversarial risk of $c_1\eta d$ is possible for any hypothesis class with VC-dimension $d$, whereas obtaining $\eta$-adversarial risk of $c_2\eta d$ can't be achieved for any hypothesis class with VC-dimension $d$. However, there is a large gap between $c_1, c_2$. Can we close or shrink this gap?

- **Finding the correct multiplicative factor in the agnostic case.** Our results show that in the agnostic case, there must be a constant $C \geq 2$ so that the best adversarial risk attainable is $C \cdot \mathsf{OPT}$. What is the value of $C$?

- **Characterizing proper robust learning.** In the proper and realizable case, our stable learner for linear classifiers depends on $d^3$, while our lower bound depends linearly on $d$, as in the general improper case. It remains open to identify the correct dependence on $d$.

- **Characterizing the role of randomness.** Our impossibility result for the agnostic learning (Theorem 4.2) only applies to deterministic learners. It remains open to either effectively use randomness during the learning (known or unknown to the adversary) and obtain an agnostic learner, or to extend the negative result to cover such randomized learners as well.

## Acknowledgments

Amin Karbasi acknowledges funding in direct support of this work from NSF (IIS-1845032), ONR (N00014- 19-1-2406), and the AI Institute for Learning-Enabled Optimization at Scale (TILOS). Mohammad Mahmoody is supported by NSF grants CCF-1910681 and CNS1936799. Shay Moran is a Robert J. Shillman Fellow; he acknowledges support by ISF grant 1225/20, by BSF grant 2018385, by an Azrieli Faculty Fellowship, by Israel PBC-VATAT, by the Technion Center for Machine Learning and Intelligent Systems (MLIS), and by the the European Union (ERC, GENERALIZATION, 101039692). Views and opinions expressed are however those of the author(s) only and do not necessarily reflect those of the European Union or the European Research Council Executive Agency. Neither the European Union nor the granting authority can be held responsible for them. We thank anonymous NeurIPS 2022 reviewers for helping us to improve this paper, and for pointing out good motivating examples.

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
