# Supplementary Material

## A  Proof of Theorem 3.1 (Realizable Case – Positive Result)

**Theorem** (Restatement of Theorem 3.1). *There exists a constant $c_1 > 0$ so that the following holds. Let $\mathcal{H}$ be a hypothesis class with VC dimension $d$ and let $\eta \in (0, 1)$. Then there exists a learner $\mathsf{Lrn}$ having $\eta$-adversarial risk*

$$\varepsilon_n^{\mathsf{Adv}}(\mathsf{Lrn}|D, \eta) \leq c_1 \eta d$$

*for any distribution $D$ realizable by $\mathcal{H}$ and for any sample size $n \geq 1/\eta$.*

To prove Theorem 3.1, we will use the STABLE PARTITION AND VOTE (or SPV for short) meta learner described in Figure 1 with the One-inclusion graph algorithm of Haussler, Littlestone, and Warmuth [1994] as the input learner. First, we prove a more general result on the performance of our SPV meta learner. We denote the algorithm obtained by executing SPV with a learner $\mathsf{Lrn}$ as the input algorithm by $\mathsf{SPV}(\mathsf{Lrn})$.

**Lemma A.1** (General performance of SPV). *Let $\mathcal{H}$ be a concept class, $D$ be a distribution over examples, and $\mathsf{Lrn}$ be a learning rule. Let also $\eta \in (0, 1)$ be the stability parameter given to SPV and let $n \geq 1/\eta$ be the sample size. Then $\mathsf{SPV}(\mathsf{Lrn})$ has $\eta$-adversarial risk*

$$\varepsilon_n^{\mathsf{Adv}}(\mathsf{SPV}(\mathsf{Lrn})|D, \eta) \leq 6\varepsilon_{\lceil 1/(7\eta) \rceil}(\mathsf{Lrn}|D).$$

*Recall that $\varepsilon_{\lceil 1/(7\eta) \rceil}(\mathsf{Lrn}|D)$ is the expected population loss of $\mathsf{Lrn}$ when trained on a sample of size $\lceil 1/(7\eta) \rceil$ from $D$ (in the standard, non adversarial, setting).*

*Proof.* Let $S \sim D^n$ be the input sample, and $(x, y) \sim D$ be the test example. Note that for all $i \in [t]$ (where $t = \lfloor 7\eta n \rfloor$ is the number of subsamples of size at least $\frac{1}{7\eta}$ in the partition made by SPV) it holds that $\mathbb{E}\big[1[h_i(x) \neq y]\big] \leq \varepsilon_{\lceil 1/(7\eta) \rceil}(\mathsf{Lrn}|D)$. By applying linearity of expectation we get

$$\mathbb{E}\left[\frac{1}{t}\sum_{i=1}^{t} 1[h_i(x) \neq y]\right] \leq \varepsilon_{\lceil 1/(7\eta) \rceil}(\mathsf{Lrn}|D).$$

By Markov's inequality:

$$\Pr\left[\frac{1}{t}\sum_{i=1}^{t} 1[h_i(x) \neq y] \geq 1/6\right] \leq 6\varepsilon_{\lceil 1/(7\eta) \rceil}(\mathsf{Lrn}|D).$$

Let $S' \in B_\eta(S)$. Let $h' = \mathsf{SPV}(\mathsf{Lrn})(S')$, and for all $i \in [t]$ let $h_i'$ be the hypothesis obtained by training $\mathsf{Lrn}$ on $S'^{(i)}$. Note that, since $S$ and $S'$ are $\eta$-close by, and since $n \geq 1/\eta$ it holds that

$$\frac{1}{t}\sum_{i=1}^{t} 1\left[S^{(i)} \neq S'^{(i)}\right] \leq \frac{\eta n}{\lfloor 7\eta n \rfloor} \leq 1/6.$$

Hence it is implied that $\frac{1}{t}\sum_{i=1}^{t} 1\left[h_i(x) \neq h_i'(x)\right] \leq 1/6$. Thus, the event that $\frac{1}{t}\sum_{i=1}^{t} 1[h_i'(x) \neq y] \geq 1/3$ implies (or, is contained in) the event that $\frac{1}{t}\sum_{i=1}^{t} 1[h_i(x) \neq y] \geq 1/6$, hence,

$$\Pr\left[\frac{1}{t}\sum_{i=1}^{t} 1[h_i'(x) \neq y] \geq 1/3\right] \leq 6\varepsilon_{\lceil 1/(7\eta) \rceil}(\mathsf{Lrn}|D).$$

Since $h'(x)$ is a majority vote of $\{h_1'(x), \ldots, h_t'(x)\}$, the above implies that

$$\Pr[h'(x) \neq y] \leq 6\varepsilon_{\lceil 1/(7\eta) \rceil}(\mathsf{Lrn}|D).$$

Since $S'$ is an arbitrary sample in $B_\eta(S)$, the above implies that $\mathsf{SPV}(\mathsf{Lrn})$ has the stated $\eta$-adversarial risk. $\qquad\square$

To prove Theorem 3.1, we will need an optimal learner as an input learner for SPV.

**Theorem A.2** (Haussler, Littlestone, and Warmuth [1994]). *Let $\mathcal{H}$ be a concept class with VC-dimension $d$, and let $D$ be a distribution realizable by $\mathcal{H}$. Let also $n \in \mathbb{N}$, and let $\mathsf{Lrn}$ be the One-inclusion graph algorithm. Then $\varepsilon_n(\mathsf{Lrn}|D) \leq \frac{d}{n+1}$.*

Theorem 3.1 can now be immediately inferred as a direct application of Lemma A.1 and Theorem A.2.

**Corollary A.3** (Realizable case – positive result). *Let $\mathcal{H}$ be a concept class with VC-dimension $d$, let $D$ be a distribution realizable by $\mathcal{H}$, and let $\mathsf{Lrn}$ be the One-inclusion graph algorithm. Let also $\eta \in (0,1)$ be the stability parameter given to $\mathsf{SPV}$ and let $n \geq 1/\eta$ be the sample size. Then $\mathsf{SPV}(\mathsf{Lrn})$ has $\eta$-adversarial risk*

$$\varepsilon_n^{\mathsf{Adv}}(\mathsf{SPV}(\mathsf{Lrn})|D, \eta) \leq 42\eta d.$$

*Proof.* By Theorem A.2, plug in $\varepsilon_{\lceil 1/(7\eta)\rceil}(\mathsf{Lrn}|D) \leq \frac{d}{\lceil 1/(7\eta)\rceil+1} \leq 7\eta d$ to Lemma A.1 and the result follows. $\square$

# B  Proof of Theorem 3.3 (Realizable Case – Impossibility Result)

**Randomized Learning Rules.** The impossibility result in Theorem 3.3 extends to randomized learning rules. But in order for the statement in Theorem 3.3 to be meaningful, we need to define adversarial risk with respect to randomized learners. As common in the literature on learning theory (see, e.g. the book of Shalev-Shwartz and Ben-David [2014]) we model randomized learners as deterministic learning rules with *continuous* predictions $p \in [0,1]$, and loss function $\ell(p,y) = |p - y|$. Indeed, the loss of a deterministic learner predicting a value $p \in [0,1]$ under the loss function $|y - p|$ is equal to the expected $0/1$-loss of a randomized learner predicting $1$ with probability $p$. In the course of discussing the impossibility result, a *learning algorithm* $\mathsf{Lrn} \colon (\mathcal{X} \times \{0,1\})^* \to [0,1]^{\mathcal{X}}$ is a deterministic mapping which takes an input sample $S \in (\mathcal{X} \times \{0,1\})^*$ and maps it to a hypothesis $f \in [0,1]^{\mathcal{X}}$. We re-define $\eta$-adversarial risk with this view of randomized learners as *randomized $\eta$-adversarial risk*.

**Definition B.1** (Randomized $\eta$-Adversarial Risk). Let $\eta \in (0,1)$ be the adversaries' budget, let $\mathsf{Lrn}$ be a learning rule, and let $D$ be a distribution over examples. The *randomized $\eta$-adversarial risk* of $\mathsf{Lrn}$ w.r.t $D$ and sample size $n$ is defined by

$$\varepsilon_n^{\mathsf{Adv}}(\mathsf{Lrn}|D, \eta) := \mathbb{E}_{S \sim D^n, (x,y) \sim D} \left[ \sup_{S' \in B_\eta(S)} |\mathsf{Lrn}(S')(x) - y| \right].$$

The above definition of adversarial risk captures the case of an adversary that knows the expected prediction of the learner (that is, its test-time randomness), but not the learner's "internal" randomness (computation-time randomness). Indeed, the supremum is taken only with respect to the expected prediction, and not with respect to a specific execution of the algorithm determined by its internal randomness. Note that deterministic learners are a special case ($\{0,1\}$-valued outputs), in which case this definition collapses to the previous Definition 2.2. To avoid further notation, note that we overloaded the notation $\varepsilon_n^{\mathsf{Adv}}$ from Definition 2.2 in the above more general definition.

We are now ready to prove the impossibility result.

**Theorem** (Restatement of Theorem 3.3). *There exists a constant $c_2 > 0$ so that the following holds. Let $\mathcal{H}$ be a non-trivial hypothesis class with VC dimension $d$ and let $\eta \in (0,1)$. Then, there exists a distribution $D$ realizable by $\mathcal{H}$, so that every learner $\mathsf{Lrn}$ has $\eta$-adversarial risk*

$$\varepsilon_n^{\mathsf{Adv}}(\mathsf{Lrn}|D, \eta) \geq \min\{c_2 \eta d, 1/100\}$$

*for any sample size $n \geq 1/\eta$.*

*Proof.* Let $\mathcal{H}$ be a non-trivial concept class; in particular this means that its VC-dimension $d$ satisfies $d \geq 1$. Let $\eta \in (0,1)$ be the adversaries' budget and let $\mathsf{Lrn}$ be an arbitrary learner. We need to show that there exists a distribution $D$ realizable by $\mathcal{H}$ so that $\varepsilon_n^{\mathsf{Adv}}(\mathsf{Lrn}|D, \eta) \geq \min\{\eta d/32, 1/100\}$.

It suffices to consider the case when $\eta d/32 \leq 1/100$ and prove that $\varepsilon_n^{\mathsf{Adv}}(\mathsf{Lrn}|D, \eta) \geq \eta d/32$. Indeed, in the complementing case we have $\eta d/32 > 1/100$ and we need to show that $\varepsilon_n^{\mathsf{Adv}}(\mathsf{Lrn}|D, \eta) \geq$

$1/100$. Notice that $\eta d/32 > 1/100$ is equivalent to $\eta > \frac{32}{100d}$, and thus it suffices to show that even if the adversary's budget $\eta$ is reduced to $\eta = \frac{32}{100d}$ then $\varepsilon_n^{\mathsf{Adv}}(\mathsf{Lrn}|D,\eta) \geq 1/100$. The latter indeed follows from the case when $\eta d/32 \leq 1/100$, because $\frac{32}{100d} \cdot d/32 = 1/100$.

We thus assume that $\eta d/32 \leq 1/100$ and set out to prove that $\varepsilon_n^{\mathsf{Adv}}(\mathsf{Lrn}|D,\eta) \geq \eta d/32$. We first consider the case when the VC-dimension of $\mathcal{H}$ is $d \geq 2$ and later handle the case when $d = 1$.

**The VC dimensions is** $d \geq 2$. Let $V = \{v_1, \ldots, v_d\} \subset \mathcal{X}$ be shattered by $\mathcal{H}$. Define a distribution $D_{\mathcal{X}}$ over $V$ as follows. Set $D_{\mathcal{X}}(v_i) = \eta/2$ for all $2 \leq i \leq d$, and set $D_{\mathcal{X}}(v_1) = 1 - \eta(d-1)/2$. Notice that $D_{\mathcal{X}}$ is well defined since $d \geq 2$ and $\eta \leq 2/(d-1)$ (the latter is implied by the assumption that $\eta d/32 \leq 1/100$). For any labeling function $\ell \in \mathcal{Y}^V$, let $D_\ell$ denote the distribution over examples defined by $D_\ell(v_i, \ell(v_i)) = D_{\mathcal{X}}(v_i)$ for all $i \in [d]$. Note that $D_\ell$ is realizable, since $V$ is shattered. It suffices to show that if the label vector $\ell \sim \mathcal{Y}^V$ is drawn uniformly at random then

$$\mathbb{E}_{\ell \sim \mathcal{Y}^V} \mathbb{E}_{S \sim D_\ell^n, (x,y) \sim D_\ell} \left[ \sup_{S' \in B_\eta(S)} |\mathsf{Lrn}(S')(x) - y| \right] \geq \eta(d-1)/16. \tag{1}$$

Indeed, the above implies that there exists $\ell \in \mathcal{Y}^V$ such that

$$\mathbb{E}_{S \sim D_\ell^n, (x,y) \sim D_\ell} \left[ \sup_{S' \in B_\eta(S)} |\mathsf{Lrn}(S')(x) - y| \right] \geq \eta(d-1)/16$$

$$\geq \eta d/32. \tag{$d \geq 2$}$$

We establish Equation 1 in two steps:

1. For a sample $S$ let $S^u$ be the unlabeled input sample underlying it. We say that an unlabeled sample $S^u$ and an instance $x$ are *hard* if $x \neq v_1$ and $x$ appears at most $\eta n$ times in $S^u$. In the first step we show that $\Pr_{\ell, S, (x,y)}[S^u, x \text{ are hard}] \geq \eta(d-1)/4$.

2. Let $E_2$ denote the event of all label vectors $\ell$, input samples $S$, and test examples $(x,y)$ such that $\sup_{S' \in B_\eta(S)} |\mathsf{Lrn}(S')(x) - y| \geq 1/2$. In the second step we show that $\Pr[E_2 | S^u, x \text{ are hard}] \geq 1/2$.

Indeed, once we prove both steps we have:

$$\mathbb{E}_{l \sim \mathcal{Y}^V} \mathbb{E}_{S \sim D_\ell^n, (x,y) \sim D_\ell} \left[ \sup_{S' \in B_\eta(S)} |\mathsf{Lrn}(S')(x) - y| \right] \geq \frac{1}{2} \cdot \Pr[E_2]$$

$$\geq \frac{1}{2} \cdot \Pr[S^u, x \text{ are hard}] \cdot \Pr[E_2 | S^u, x \text{ are hard}]$$

$$\geq \frac{1}{2} \cdot \frac{\eta(d-1)}{4} \cdot \frac{1}{2} = \eta(d-1)/16,$$

as desired.

Let us prove step 1. Notice that $S^u$ and $x$ are distributed according to the marginal distribution $D_{\mathcal{X}}^{n+1}$. Thus, $x \neq v_1$ with probability $\eta(d-1)/2$, and given that $x \neq v_1$ the expected number of appearances of $x$ in $S^u$ is $\eta n/2$. Therefore, by Markov's inequality, the probability that $S^u$ and $x$ are hard given that $x \neq v_1$ is at least $\frac{\eta n/2}{\eta n} = 1/2$. Thus, the overall probability that $S^u, x$ are hard is at least $\eta(d-1)/4$.

We now prove step 2. Let $S^u, x$ be hard. It suffices to show that

$$\mathbb{E}_{\ell(x_1), \ldots, \ell(x_n), y} \left[ \sup_{S' \in B_\eta(S)} |\mathsf{Lrn}(S')(x) - y| \,\Big|\, S^u, x \right] \geq \frac{1}{2},$$

where $\ell(x_i)$ is the label of the $i$'th instance in $S^u$ and $y$ is the test label. Crucially, notice that the test-label $y$ is independent of $S^u$, $x$, and all other labels $\ell(x_i)$ for $x_i \in S^u$ such that $x_i \neq x$. Thus, even conditioned on $S^u, x$ and all labels of $x_i \neq x$, the test-label $y$ is distributed uniformly in $\mathcal{Y} = \{0, 1\}$.

Define samples $S'_0, S'_1$ to be the same as $S'$ with the exception that every appearance of $x$ in $S'_0$ is labeled with 0 in $S'_0$ and with 1 in $S'_1$. Note that both $S'_0, S'_1 \in B_\eta(S)$, because $S^u, x$ are hard. We claim that, with probability at least half over the drawing of the $\ell(x_i)$'s and $y$ we have

$$|\mathsf{Lrn}(S'_0)(x) - \ell(y)| \geq 1/2 \quad \text{or} \quad |\mathsf{Lrn}(S'_1)(x) - \ell(y)| \geq 1/2.$$

Having this in hand, and given that $\hat{S}$ is hard, we are done: both $S'_0, S'_1 \in B_\eta(S)$, and Item 2 follows.

It thus remains to show that indeed $|\mathsf{Lrn}(S'_0)(x) - y| \geq 1/2$ or $|\mathsf{Lrn}(S'_1)(x) - \ell(y)| \geq 1/2$ with probability at least $1/2$ over the drawing of the $\ell(x_i)$'s and $y$. This is achieved by a simple case analysis:

- if both $\mathsf{Lrn}(S'_0)(x), \mathsf{Lrn}(S'_1)(x) \leq 1/2$ then with probability $1/2$ we have $y = 1$ and the claim follows. The case $\mathsf{Lrn}(S'_0)(x), \mathsf{Lrn}(S'_1)(x) > 1/2$ is treated similarly.

- If $\mathsf{Lrn}(S'_0)(x) \leq 1/2, \mathsf{Lrn}(S'_1)(x) \geq 1/2$ then $|\mathsf{Lrn}(S'_0)(x) - y| \geq 1/2$ or $|\mathsf{Lrn}(S'_1)(x) - y| \geq 1/2$ with probability 1 and the claim follows. The case $\mathsf{Lrn}(S'_0)(x) > 1/2, \mathsf{Lrn}(S'_1)(x) < 1/2$ is treated similarly.

This finishes the proof of Theorem 3.3 when the VC-dimension $d$ is at least 2.

**The VC-dimension is $d = 1$.** In this case, we can not define the distribution $D_{\mathcal{X}}$ as before because $d < 2$. However, the fact that $\mathcal{H}$ is non-trivial allows to modify the definition as follows. Let $x_1, x_2 \in \mathcal{X}$ and $h_1, h_2 \in \mathcal{H}$ so that $h_1(x_1) = h_2(x_1)$ and $h_1(x_2) \neq h_2(x_2)$, guaranteed by the fact that $\mathcal{H}$ is non-trivial. Set $V = \{x_1, x_2\}$, and define the distribution $D_{\mathcal{X}}$ by $D_{\mathcal{X}}(x_1) = 1 - \eta/2, D_{\mathcal{X}}(x_2) = \eta/2$ as in the case $d \geq 2$. Also, define the random labeling function $\ell$ to agree with $h_1$ on with probability half and with $h_2$ with probability half. The rest of the proof is the same. $\square$

## C  Proof of Theorem 3.6 (Realizable and Proper Case – Positive Result)

**Theorem** (Restatement of Theorem 3.6)**.** *There exists a constant $c > 0$ so that the following holds. Let $\mathcal{H}$ be the class of halfspaces over $\mathbb{R}^d$ for some $d \geq 1$, and let $\eta \in (0, 1)$. Then, there exists a proper learner $\mathsf{Lrn}$ having $\eta$-adversarial risk*

$$\varepsilon_n^{\mathsf{Adv}}(\mathsf{Lrn}|D, \eta) \leq c\eta d^3$$

*for any distribution $D$ realizable by $\mathcal{H}$ and for any sample size $n \geq 1/\eta$.*

To derive Theorem 3.6, we reinforce the SPV algorithm with a technique introduced by Kane, Livni, Moran, and Yehudayoff [2019] and further developed by Bousquet, Hanneke, Moran, and Zhivotovskiy [2020]. This technique allows in certain cases to *project* a majority vote of hypotheses from the class $\mathcal{H}$ back to $\mathcal{H}$. Its applicability hinges on a combinatorial parameter called the *projection number*:

**Definition C.1** (Projection Number)**.** Let $\mathcal{H}$ be a concept class. For any $\ell \geq 2$ and for any multiset $\mathcal{H}' \subset \mathcal{H}$ define the set $\mathcal{X}_{\mathcal{H}', \ell}$ to be the set of all $x \in \mathcal{X}$, for which the number of hypotheses in $\mathcal{H}'$ that disagree with $\mathsf{Maj}(\mathcal{H}')(x)$ is less than $|\mathcal{H}'|/\ell$. The Projection Number of the class $\mathcal{H}$, denoted $k_p = k_p(\mathcal{H})$, is defined to be the smallest $\ell$ so that for any finite multiset $\mathcal{H}' \subset \mathcal{H}$, there exist $h \in \mathcal{H}$ such that $h(x) = \mathsf{Maj}(\mathcal{H}')(x)$ for all $x \in \mathcal{X}_{\mathcal{H}', \ell}$. If no such $\ell$ exists then $k_p = \infty$.

First, let us analyze the general performance of PSPV.

**Lemma C.2** (General performance of PSPV)**.** *Let $\mathcal{H}$ be a concept class with a finite projection number $k_p < \infty$. Let $D$ be a distribution over examples, and let $\mathsf{Lrn}_p$ be a proper learning rule. Let also $\eta \in (0, 1)$ be the stability parameter given to PSPV and let $n \geq 1/\eta$ be the sample size. Then PSPV($\mathsf{Lrn}_p$) is a proper learning rule having $\eta$-adversarial risk*

$$\varepsilon_n^{\mathsf{Adv}}(\mathsf{PSPV}(\mathsf{Lrn}_p)|D, \eta) \leq 4k_p \varepsilon_{\lceil 1/(5k_p\eta) \rceil}(\mathsf{Lrn}_p|D).$$

*Proof.* The proof follows the same lines as the proof of Lemma A.1. Let $S \sim D^n$ be the input sample, and $(x, y) \sim D$ be the test example. Note that for all $i \in [t]$ (where $t = \lfloor 5k_p\eta n \rfloor$

is the number of subsamples of size at least $\frac{1}{5k_p\eta}$ in the partition made by PSPV) it holds that $\mathbb{E}\big[1[h_i(x) \neq y]\big] \leq \varepsilon_{\lceil 1/(5k_p\eta)\rceil}(\mathsf{Lrn}_p|D)$. By applying linearity of expectation we get

$$\mathbb{E}\left[\frac{1}{t}\sum_{i=1}^{t} 1[h_i(x) \neq y]\right] \leq \varepsilon_{\lceil 1/(5k_p\eta)\rceil}(\mathsf{Lrn}_p|D).$$

By Markov's inequality:

$$\Pr\left[\frac{1}{t}\sum_{i=1}^{t} 1[h_i(x) \neq y] \geq \frac{1}{4k_p}\right] \leq 4k_p\varepsilon_{\lceil 1/(5k_p\eta)\rceil}(\mathsf{Lrn}_p|D).$$

Let $S' \in B_\eta(S)$. Let $h' = \mathsf{PSPV}(\mathsf{Lrn}_p)(S')$, and for all $i \in [t]$ let $h'_i$ be the hypothesis obtained by training $\mathsf{Lrn}_p$ on $S'^{(i)}$. Note that, since $S$ and $S'$ are $\eta$-close by, and since $n \geq 1/\eta$ it holds that

$$\frac{1}{t}\sum_{i=1}^{t} 1\left[S^{(i)} \neq S'^{(i)}\right] \leq \frac{\eta n}{\lfloor 5k_p\eta n\rfloor} \leq \frac{1}{4k_p}.$$

Hence it is implied that $\frac{1}{t}\sum_{i=1}^{t} 1\left[h_i(x) \neq h'_i(x)\right] \leq \frac{1}{4k_p}$. Thus, the event that $\frac{1}{t}\sum_{i=1}^{t} 1[h'_i(x) \neq y] \geq \frac{1}{2k_p}$ implies (or, is contained in) the event that $\sum_{i=1}^{t} 1[h_i(x) \neq y] \geq \frac{1}{4k_p}$, hence:

$$\Pr\left[\frac{1}{t}\sum_{i=1}^{t} 1[h'_i(x) \neq y] \geq \frac{1}{2k_p}\right] \leq 4k_p\varepsilon_{\lceil 1/(5k_p\eta)\rceil}(\mathsf{Lrn}_p|D).$$

Note that by definition of projection number it holds that the hypothesis $h' \in \mathcal{H}$ returned by the algorithm exists. Hence, by definition of $\mathcal{X}_{\{h'_1,\ldots,h'_t\},2k_p}$ the above implies that

$$\Pr[h'(x) \neq y] \leq 4k_p\varepsilon_{\lceil 1/(5k_p\eta)\rceil}(\mathsf{Lrn}_p|D).$$

Since $S'$ is an arbitrary sample in $B_\eta(S)$, the above implies that $\mathsf{PSPV}(\mathsf{Lrn}_p)$ has the stated $\eta$-adversarial risk. $\qquad\square$

To prove Theorem 3.6 we will use the following result regarding the projection number of halfspaces.

**Theorem C.3** (Kane, Livni, Moran, and Yehudayoff [2019], Braverman, Kol, Moran, and Saxena [2019], Bousquet, Hanneke, Moran, and Zhivotovskiy [2020]). *Let $\mathcal{H}$ be the class of halfspaces over $\mathbb{R}^m$. Then $k_p(\mathcal{H}) = d(\mathcal{H}) = m + 1$.*

We will use the SVM learner as an input learner for PSPV.

**Theorem C.4** (Vapnik and Chervonenkis [1974]). *Let $m \geq 1$ and let $\mathcal{H}$ be the class of halfspaces over $\mathbb{R}^m$. Let $D$ be a distribution realizable by $\mathcal{H}$. Let also $n \in \mathbb{N}$, and let $\mathsf{Lrn}_p$ be the SVM algorithm. Then $\varepsilon_n(\mathsf{Lrn}_p|D) \leq \frac{m+1}{n+1}$.*

Theorem 3.6 now follows as an immediate application of Theorem C.3, Theorem C.4 and Lemma C.2.

**Corollary C.5** (Realizable and proper case – positive result). *Let $m \geq 1$, let $\mathcal{H}$ be the class of halfspaces over $\mathbb{R}^m$, and let $d = m + 1$ be the VC-dimension of $\mathcal{H}$. Let $D$ be a distribution realizable by $\mathcal{H}$, and let $\mathsf{Lrn}_p$ be the SVM learner. Let also $\eta \in (0, 1)$ be the stability parameter given to $\mathsf{PSPV}$ and let $n \geq 1/\eta$ be the sample size. Then $\mathsf{PSPV}(\mathsf{Lrn}_p)$ has $\eta$-adversarial risk*

$$\varepsilon_n^{\mathsf{Adv}}(\mathsf{PSPV}(\mathsf{Lrn}_p)|D,\eta) \leq 20\eta d^3.$$

*Proof.* By Theorem C.3, if $\mathcal{H}$ is the class of halfspaces over $\mathbb{R}^m$ then its projection number is $k_p = d = m + 1$. Also, by Theorem C.4, we have that $\varepsilon_{\lceil 1/(5d\eta)\rceil}(\mathsf{Lrn}_p|D) \leq 5\eta d^2$. Plug both results to Lemma C.2, and the result follows. $\qquad\square$

# D   Proof of Theorem 4.1 (Agnostic Case – Positive Result)

**Theorem** (Restatement of Theorem 4.1). *There exist constants $c_1, c_2$ so that the following holds. Let $\mathcal{H}$ be a hypothesis class with VC dimension $d$ and let $\eta \in (0, 1)$. Then, there exists a learner $\mathsf{Lrn}$ having $\eta$-adversarial risk*

$$\varepsilon_n^{\mathsf{Adv}}(\mathsf{Lrn}|D, \eta) \leq c_2 \cdot \mathsf{OPT} + c_1 \cdot d \cdot \eta$$

*for any distribution $D$ over examples and for any sample size $n \geq 1/\eta$.*

To derive Theorem 4.1, we use an agnostic variation of the One-inclusion graph learner.

**Theorem D.1** (Corollary of Lemma 16 in [Long, 1999]). *There exists a constant $C$ such that the following holds. Let $\mathcal{H}$ be a concept class with VC-dimension $d$ and let $\mathsf{Lrn}$ be the agnostic variation of the One-inclusion graph algorithm implied by Lemma 16 in [Long, 1999]. Let also $n$ be the sample size. Then, for any distribution $D$ over examples (not necessarily such that is realizable by $\mathcal{H}$), it holds that $\varepsilon_n(\mathsf{Lrn}|D) \leq C(\mathsf{OPT} + d/n)$.*

Theorem 4.1 is implied by the following immediate corollary of Theorem 4.1 and Lemma A.1.

**Corollary D.2** (Agnostic case – positive result). *There exists a constant $C$ such that the following holds. Let $\mathcal{H}$ be a concept class with $VC$ dimension $d$, let $\eta \in (0, 1)$ be the stability parameter given to $\mathsf{SPV}$, and let $D$ be a (not necessarily realizable) distribution over examples. Let also $n \geq 1/\eta$ be the sample size. Then $\mathsf{SPV}(\mathsf{Lrn})$ has $\eta$-adversarial risk*

$$\varepsilon_n^{\mathsf{Adv}}(\mathsf{SPV}(\mathsf{Lrn})|D, \eta) \leq 6C\mathsf{OPT} + 42C\eta d,$$

*where $\mathsf{Lrn}$ is the agnostic variant of the One-inclusion graph algorithm mentioned in Theorem D.1.*

*Proof.* By Theorem D.1, there exists a constant $C$ such that $\varepsilon_{\lceil 1/(7\eta) \rceil}(\mathsf{Lrn}|D) \leq C\mathsf{OPT} + 7C\eta d$. Plug this into Lemma A.1 and the result follows. $\square$

# E   Proof of Theorem 4.2 (Agnostic Case – Impossibility Result)

**Theorem** (Restatement of Theorem 4.2). *Let $\eta' \in (0, 1), n \in \mathbb{N}$. For any hypothesis class $\mathcal{H}$ that has at least two hypotheses and. for any deterministic learner, there is a distribution $D$ over (two) examples and $\eta = \eta' + \widetilde{O}(1/\sqrt{n})$ such that $\mathsf{Lrn}$ has $\eta$-adversarial risk*

$$\varepsilon_n^{\mathsf{Adv}}(\mathsf{Lrn}|D, \eta) \geq 2\mathsf{OPT} + \Omega(\eta') - O(1/n).$$

Let $h_1, h_2 \in \mathcal{H}$ be two distinct hypotheses and let $x \in \mathcal{X}$ such that $h_1(x) \neq h_2(x)$. In this proof we consider distributions $D$ supported only on $\{(x, 0), (x, 1)\}$. Notice that such a distribution is determined by the probability $p = \Pr_{(x,y)\sim D}[y = 1]$ and hence can be thought of as a coin with bias $p$. Thus, the task of agnostic learning such distributions with respect to instance-targeted data poisoning boils down to predicting a random $p$-coin toss given an input sample of $n$ $p$-coin tosses out of which at most $\eta \cdot n$ tosses are flipped by an adversary who *knows* the result of the coin toss that needs to be predicted. We summarize this in the following game:

**Definition E.1** (The coin game). The coin game is parameterized by $(n, \eta)$ where $n \in \mathbb{N}, \eta \in (0, 1)$, and the game is played between an adversary $\mathsf{Adv}$ and a learner $\mathsf{Lrn}$ as follows.

1. $\mathsf{Adv}$ picks $p \in [0, 1]$.

2. $c_1, \ldots, c_{n+1} \sim X_p^{n+1}$, where $X_p$ is a binary random variable satisfying $\Pr[X_p = 1] = p$.

3. $\mathsf{Adv}$ changes $\overline{c} = (c_1, \ldots, c_n)$ into $\overline{c}' = (c_1', \ldots, c_n')$ where $\mathsf{d}_{\mathsf{H}}(\overline{c}, \overline{c}') \leq \eta \cdot n$.

4. $\mathsf{Lrn}$ gets to see $\overline{c}' = (c_1', \ldots, c_n')$ and outputs a bit $c \in \{0, 1\}$.

5. $\mathsf{Lrn}$ wins if $c = c_{n+1}$, and $\mathsf{Adv}$ wins otherwise.

In this game, we define $\mathsf{OPT}_p = \min\{p, 1 - p\}$ to be the optimal error of the learner if it had known $p$, and we define $\mathsf{ERR} = \Pr[c \neq c_{n+1}]$ (over all the randomness involved) to be the *error* of the game (i.e., when the learner does not win). We also refer to $\mathsf{ERR} - \mathsf{OPT}_p$ as the regret.[4]

---

[4]Note that $\mathsf{OPT}_p$ is a random variable in general, if the adversary is randomized. But if the adversary uses a deterministic strategy for the fixed $p$, then $\mathsf{OPT}_p$ is a constant.

**Theorem E.2.** *For any $\eta' \in [0, 1/2]$ and any deterministic learner* Lrn *that participates in the coin game of Definition E.1, there is an adversary* Adv *with a fixed choice of $p$ (determining* $\mathsf{OPT} = \mathsf{OPT}_p$*) and $\eta = \eta' + \widetilde{O}(1/\sqrt{n})$ such that when we run the game of Definition E.1 with parameters $(n, \eta)$, it holds that* $\mathsf{ERR} - \mathsf{OPT} \geq 1/2 + \eta' - O(1/n)$.

**Remark 1** (On deterministic adversaries). *In Theorem E.2 we show the existence of an adversary with a fixed choice of $p$. This adversary is in fact randomized. Here we remark that, for every fixed (even randomized) learner* Lrn *and a fixed choice of $p$, there is always a deterministic adversary that achieves the maximum regret (for such* Lrn, $p$*). The reason is that if by using randomness $r_{\mathsf{Adv}}$ the adversary achieves expected regret $R(r_{\mathsf{Adv}})$ over the randomness of the learner, then its overall regret will be $\mathbb{E}_{r_{\mathsf{Adv}}}[R(r_{\mathsf{Adv}})]$. Therefore, if $r_{\mathsf{Adv}}^{(p)}$ is the randomness (for fixed $p$) that maximizes $R(r_{\mathsf{Adv}})$, the adversary can simply fix its randomness to $r_{\mathsf{Adv}}^{(p)}$ without decreasing its gain. This means that without loss of generality, the adversary of Theorem E.2 is deterministic. In addition, since the adversary sends the first message $p$, the overall optimal strategy* Adv *(who picks $p$ potentially in a randomized way) can also fix $p$ to what maximizes $R(r_{\mathsf{Adv}}^{(p)})$, which makes* Adv *fully deterministic.*

**Deriving Theorem 4.2.** We first show how to derive Theorem 4.2 from Theorem E.2.

*Proof of Theorem 4.2.* First assume $\eta' \leq 1/2$, and at the end we explain how to deal with $\eta' > 1/2$. By Theorem E.2, there is an adversary (with a fixed choice of $p$) in the coin game of Definition E.1 such that $\mathsf{ERR} - \mathsf{OPT} \geq 1/2 + \eta' - O(1/n)$ when we use $(n, \eta)$ as game parameters. Since $\mathsf{ERR} \leq 1$, we have $\mathsf{OPT} \leq 1/2 - \eta' + O(1/n)$, and so

$$\mathsf{ERR} - \mathsf{OPT} \geq 1/2 + \eta' - O(1/n) \geq \mathsf{OPT} + 2\eta' - O(1/n).$$

This implies that $\mathsf{ERR} \geq 2\mathsf{OPT} + 2\eta' - O(1/n)$. Note that $\mathsf{OPT}$ is indeed the minimal error that the learner can achieve by outputting any of the constant coins $0, 1$, which in turn refers to outputting either of $h_0, h_1$ from the hypothesis class. In addition, $\mathsf{ERR}$ is equal to the adversarial risk for parameters $n, \eta$ and the distribution $D_p$ for this particular attack. This means that

$$\varepsilon_n^{\mathsf{Adv}}(\mathsf{Lrn}|D, \eta) \geq 2\mathsf{OPT} + 2\eta' - O(1/n),$$

which implies Theorem 4.2. Now, if $\eta' > 1/2$, we first artificially decrease adversary's budget $\eta'$ to $\eta'' = 1/2$, which leads to

$$\varepsilon_n^{\mathsf{Adv}}(\mathsf{Lrn}|D, \eta) \geq 2\mathsf{OPT} + 2\eta'' - O(1/n),$$

but we also know that $\eta'' = \Omega(\eta')$, which again proves Theorem 4.2. $\square$

Before proving Theorem E.2 we recall two useful tools.

**Lemma E.3** (Proposition 2.1.1 in Talagrand [1995]). *Let $\mu = \mu_1 \times \ldots \mu_n$ be a product measure and $f: \mu \mapsto \{0, 1\}$ a boolean function where $\Pr[f(\mu) = 1] = 1/2$. Then, for all $b \in [n]$,*

$$\Pr_{x \sim \mu}\left[\exists x', \mathsf{d}_{\mathsf{H}}(x, x') \leq b \wedge f(x') = 1\right] \geq 1 - 2e^{-b^2/n}.$$

*In other words, with probability at least $1 - 2e^{-b^2/n}$ over the sampling of $x \sim \mu$, one can change up to $b$ of the coordinates of $x$ and obtain $x'$ (i.e., $\mathsf{d}_{\mathsf{H}}(x, x') \leq b$) such that $f(x') = 1$.*

**Lemma E.4** (Modifying coins). *Suppose $0 \leq p, p' \leq 1$, and let $q = |p - p'|$. Then there is an adversary who can change $q \cdot n$ coins, in expectation, of a sample $\overline{c} \sim X_p^n$ into $\overline{c}'$ (i.e., $\mathbb{E}[\mathsf{d}_{\mathsf{H}}(\overline{c}', \overline{c})] = q \cdot n$) such that $\overline{c}' \sim X_{p'}^n$ (Namely, the tampered sequence looks exactly like it is sampled from $X_{p'}^n$, while in reality it is being first sampled from $X_p^n$ and then modified by the adversary in $q \cdot n$ points in expectation). Moreover, the probability that the adversary changes more than $qn + \sqrt{(n \ln n)/2}$ of the coordinates is at most $1/n$.*

*Proof.* Without loss of generality, let $p' - p = q \geq 0$. Then the adversary will change each of the coins with independent probability $q$ as follows. If a coin $c_i = 1$, the adversary will not change it, which will happen with probability $p$. If $c_i = 0$, which will happen with probability $1 - p$, the adversary will change this to 1 with probability $q/(1 - p)$ over its own randomness. Note that $q = p' - p \leq (1 - p)$, and so $q/(1 - p) \in [0, 1]$ can be interpreted as a probability. The probability that $c_i = 1$ is now exactly $p + q = p'$, while the expected number of changed coins is $q \cdot n$. Finally, since the adversary's changes of the coin outcomes are done *independently* for each coin, the bound on the number of changes made by the adversary is implied by the Hoeffding-Chernoff bound. $\square$

We now prove Theorem E.2 using the two tools above.

*Proof of Theorem E.2.* Fix the deterministic learning algorithm Lrn. This means that for every given input vector $\bar{c} = (c_1, \ldots, c_n)$, we have $\mathsf{Lrn}(\bar{c}) \in \{0, 1\}$. Now define $\alpha(p) = \Pr_{\bar{c} \sim X_p^n}[\mathsf{Lrn}(\bar{c}) = 1]$.

We do a case study as follows.

- If $\alpha(0) \neq 0$, it means that $\alpha(0) = 1$ (i.e., the deterministic learner outputs 1 over the all zero vector). In this case, $\mathsf{OPT} = 0$ and $\mathsf{ERR} = 1$, which implies $\mathsf{ERR} - \mathsf{OPT} \geq 1/2 + \eta'$.

- If $\alpha(1) \neq 1$, it implies $\mathsf{ERR} - \mathsf{OPT} \geq 1/2 + \eta'$ similarly.

- If none of the above cases happens, we can assume $\alpha(b) = b$ for both $b \in \{0, 1\}$. Because the learner is deterministic, $\mathsf{Lrn}(\bar{c}) = 1$ if $\bar{c} \in \mathcal{S}$ for a fixed set $\mathcal{S} \subseteq \{0, 1\}^n$. Moreover, for all $\bar{c} \in \{0, 1\}^n$, it holds that $\Pr[X_p^n = \bar{c}] = p^d(1-p)^{n-d}$, where $d$ is the number of non-zero coordinates of $\bar{c}$. This implies that $\alpha(p)$ is a polynomial of degree at most $n$ over $p$, which is a continuous function. Therefore, there exists $q \in (0, 1)$ such that $\alpha(q) = 1/2$. Without loss of generality, assume that $q \leq 1/2$. Then, the adversary picks $p = \max\{0, q - \eta'\}$, which guarantees $\mathsf{OPT} = p \leq 1/2 - \eta'$ (due to the assumptions $\eta', q \leq 1/2$). Then, the adversary uses Lemma E.4 to shift the coin's distribution back to $q$. For this change, the adversary makes at most $\eta' \cdot n + \sqrt{(n \ln n)/2}$ changes with probability $1 - 1/n$. We then apply the algorithm of Lemma E.3 to make further $\sqrt{n \ln(2n)}$ changes to the coins to make sure that the output of the learner is the wrong outcome (different from $c_{n+1}$) with probability $1 - 1/n$. In total, the adversary can make at most $\eta' \cdot n + \sqrt{(n \ln n)/2} + \sqrt{n \ln(2n)} \in \eta' \cdot n + \widetilde{O}(\sqrt{n})$ changes to the coin flips outcomes, while the learner's output bit is wrong with probability $1 - 1/n - 1/n = 1 - O(1/n)$. Since $\mathsf{OPT} \leq 1/2 - \eta'$ and $\mathsf{ERR} \geq 1 - O(1/n)$, we get

$$\mathsf{ERR} - \mathsf{OPT} \geq 1/2 + \eta' - O(1/n),$$

which finishes the proof. □