# OpenReview forum: "On Optimal Learning Under Targeted Data Poisoning"
_NeurIPS.cc/2022/Conference — NeurIPS 2022 Accept_

### Official Review · Reviewer_dWRe · 2022-07-07

**Rating:** 7
**Confidence:** 3
**Soundness:** 3 good
**Presentation:** 2 fair
**Contribution:** 3 good

**Summary:**

The paper information-theoretically studies the possibility and impossibility of learning from poisoned data, in a supervised learning setting with deterministic model predictions, against a relatively powerful attacker who knows the data $x$ on which the model will be tested (or deployed). Remarkably, in the realizable setting, they prove matching upper and lower bounds, up to a multiplicative constant. Namely, this error is $\Theta(\eta d)$, where $\eta$ is the fraction of poisoned data and $d$ is the VC dimension of the class of hypotheses.

In the agnostic setting, the paper also provides interesting lower and upper bounds, though the gap is then arguably larger. Namely, they prove that the error is at least (roughly) $2 OPT + \Omega(\eta) - o(1)$, for a large number of training data, while their upper bound only guarantees $\mathcal O(OPT + d \eta)$.

The authors also note that their solution in the realizable setting is not proper (the learned hypothesis aggregates the predictions of several hypotheses). In the specific case of halfspace learning, they propose a construction of a proper hypothesis, but the guarantee is significantly worse ($\mathcal O(\eta d^3)$).

**Questions:**

It would also be helpful if the authors could provide actual values for the constants, so that the readers can better understand the gap between the positive and negative results. I have seen that most of these constants are available in the Supplementary Material. I would suggest to add them to the main paper. In particular, could the authors provide the value of $c_2$ in Theorem 4.1? This constant seems especially important, as it is non-vanshing even when $\eta \rightarrow 0$.

As far as I understand, SPV, which has provable guarantees, is also very practical. In particular, it seems to have the same time-complexity as the oracle Lrn which it leverages. Is this the case? If so, I would encourage the authors to mention it in the paper (I fully appreciate the information-theoretical impossibility theorems, but it is even better if the positive theorems are computationally tractable).

It is frustrating that the algorithm used to provide the guarantee of Theorem 4.1 is not described, even informally. Would it be possible to at least provide an intuition of this algorithm?

Additionally, the time complexity of this algorithm is not discussed. Can the authors say a few words about it?

Finally, there have been lower bounds proved for Byzantine heterogeneous learning. How do these relate with the authors' lower bounds?
- https://proceedings.neurips.cc/paper/2021/hash/d2cd33e9c0236a8c2d8bd3fa91ad3acf-Abstract.html
- https://openreview.net/forum?id=jXKKDEi5vJt
- https://arxiv.org/pdf/2202.08578.pdf

**Limitations:**

It is a shame that the introduction examples that motivate the paper are unconvincing. It is indeed unclear how and why an adversary would fool autonomous cars into accelerating when shown a stop sign. It is also unclear how a loan applicant could poison a bank’s training set.

I strongly urge the authors to provide much more compelling examples, which I think are numerous. For instance, in content moderation, a troll farm employed by an authoritarian regime or a multinational firm could flag dissents’ and critics’ content on social medias to fool the moderation algorithm into considering that all such critical content should be taken down. As another example, a language model trained on “social media conversations” (like PaLM https://arxiv.org/abs/2204.02311) could be hacked by propaganda and marketing campaigns, who may want the word “meat” to be systematically associated with “delicious” rather than “not environmental-friendly”. Today, the training datasets of such algorithms are arguably full of poisonous attacks (Facebook reports removing 15 billion fake accounts every two years: https://www.techdigest.tv/2021/09/facebook-removes-15-billion-fake-accounts-in-two-years.html).

In particular, the attacker’s knowledge of $x$ can be justified in practice by the fact that, in may use cases where the model is trained on user-generated data, attackers are also users who will interact with the trained algorithm, and provide the values of $x$ that the model will process. This use case belongs to the family of backdoor attacks, which the paper would gain by mentioning.

I believe that the authors should also discuss the role of $d$. Their lower bounds essentially prove that larger models are more vulnerable to adversarial attacks. This is a very important discovery, as it strongly suggests that the race for ever larger models is also leading to significantly more vulnerable models (if we are to believe the paper).


**Strengths And Weaknesses:**

I believe that this paper is an exciting important step towards understanding the possibility and impossibility of learning. So far the literature has only provided guarantees in the case of a vanishing fraction of poisoned data. However, especially in applications like online content moderation, this assumption is very unrealistic. Arguably, as a model gains more users, while it gains more data, it also attracts more malicious users, which may thus represent an increasing fraction of the provided data. Given this, I believe that the paper provides an original and very significant contribution to the important literature on adversarial machine learning.

The paper is well-organized, but there are many typos which make the reading challenging at times. Below is a list of such typos (the list is not exhaustive):
- Line 53: “Due to adversary” => “Due to the adversary”
- Line 58: “even if the adversary knows” => “especially if the adversary knows”
- Line 66: “the Gao et al. [2021]” => “Gao et al. [2021]”
- Line 153: “Gao et al. Gao et al. [2021]” => “Gao et al. [2021]”
- Line 155: “Balcan et al. Balcan et al. [2022]” => “Balcan et al. [2022]”
- Line 156: “under instance targeted” => “under instance targeted attacks”
- Line 161: “Rosenfeld et al. Rosenfeld et al. [2020]” => “Rosenfeld et al. [2020]”
- Line 209: “depend its changes” => “make its changes depend”
- Line 212: “based on adversary’s power” => “based on the adversary’s power”
- Line 216: “In Definition B.1 present” => “In Definition B.1, we present”
Also, the word “semi-agnostic” is not defined in the paper.

Finally, the paper should clarify the difference between what they regard as “targeted attacks”, and what is sometimes known as “model-targeted attacks”, where attackers want a specific target hypothesis to be learned.
- https://proceedings.mlr.press/v139/suya21a.html
- https://arxiv.org/pdf/2202.08578.pdf (to appear in ICML’22)

EDIT after reading the authors' responses:
The authors' responses were satisfactory to me, which motivated me to increase my rating.

---

> ### Author Response · Authors · 2022-08-02
> **Answer to reviewer dWRe: Part 1**
>
> We thank the reviewer for dedicating time to read our paper and for the nice comment that “…the paper provides an original and very significant contribution to the important literature on adversarial machine learning.” Given this, we respectfully ask the reviewer to consider raising the score.
>
> We also thank the reviewer for pointing out typos (which will be fixed in a thorough proofreading of the paper), and for suggesting interesting and insightful examples (e.g. attacking a language model is a great and relevant example.) We will add more compelling examples to the next version of our paper, as suggested.
>
>
>
> *Question #1:*
>
> … the paper should clarify the difference between what they regard as “targeted attacks”, and what is sometimes known as “model-targeted attacks”…
>
> *Answer:*
>
> Thanks. The form of targeted poisoning used in our work is known as instance-targeted-poisoning (see e.g. [3,4] below). It is probably confusing that we used the shorter term “targeted poisoning” instead of “instance-targeted poisoning”. We will fix this in the next version of the paper and add a discussion surveying the different forms of targeted poisoning (including those variants mentioned by other reviewers).
>
>
> *Question #2*
>
> It would be helpful if the authors could provide actual values for the constants…
>
> *Answer:*
>
> Indeed, we did not try hard to optimize the constants in the semi-agnostic setting. As we discuss in the general reply for all reviewers, the reason is because it deviates from the main focus of the current paper and is more appropriate as an open question for future work.
> We note however that the constant $c_2$ that follows from our proof is approx. 90, and can probably be improved to ~60 at the cost of complicating the proof. (And 60 is still far from the lower bound of 2.)
> Nevertheless, we agree that it is a nicer read if we include this discussion in the main body of the paper and we will revise the paper accordingly.
>
> *Question #3:*
>
> As far as I understand, SPV, which has provable guarantees, is also very practical. In particular, it seems to have the same time-complexity as the oracle Lrn which it leverages. Is this the case? If so, I would encourage the authors to mention it in the paper (I fully appreciate the information-theoretical impossibility theorems, but it is even better if the positive theorems are computationally tractable).
>
> *Answer:*
>
> Indeed, the time complexity of SPV is roughly proportional to that of the input learner Lrn. We completely agree that this is an important comment and will add it to the manuscript. Thanks.
>
> *Question #4:*
>
> It is frustrating that the algorithm used to provide the guarantee of Theorem 4.1 is not described, even informally. Would it be possible to at least provide an intuition of this algorithm?
> Additionally, the time complexity of this algorithm is not discussed. Can the authors say a few words about it?
>
> *Answer:*
>
> Thanks for pointing this out, we will definitely try to add details (while keeping the page limit).
> The algorithm is essentially the same as SPV (which is used in the realizable case). Thus, the time complexity is the same: it is roughly proportional to that of the input learner.
>
> *Question #5:*
>
> Finally, there have been lower bounds proved for Byzantine heterogeneous learning. How do these relate with the authors' lower bounds?
>
> *Answer:*
>
>  There are at least two notable differences between our model, and the models considered in the three  papers listed in this part of the review. The first difference is that our attack model is instance-targeted (in contrast to “model-targeted”, as considered in [2]) in the sense that the adversary is only required to damage the learner’s ability to predict on a single data point known to the adversary. In the listed papers, the adversary has no knowledge of the test point. The  second difference is that in the listed papers, there are n many nodes (or “workers”), and each node has its own local distribution on the data. Note that even when overlooking the cost of communication between nodes (in order to simulate the centralized setting considered in our work), the centralized training set we get is drawn from n many (possibly) different distributions. Hence, the centralized setting is a private case of this setting (in which all local distributions are identical). So, even if the models considered in these papers were instance targeted (which they are not), a lower bound for the above distributed setting still does not imply a lower bound for the centralized model considered in our work.

---

> > ### Author Response · Authors · 2022-08-02
> > **Answer to reviewer dWRe: Part 2**
> >
> > *Question #6:*
> >
> > I believe that the authors should also discuss the role of d. Their lower bounds essentially prove that larger models are more vulnerable to adversarial attacks. This is a very important discovery, as it strongly suggests that the race for ever larger models is also leading to significantly more vulnerable models (if we are to believe the paper).
> >
> > *Answer:*
> >
> > Thanks! We agree that the dependence of the bound on d is notable. We will add a paragraph discussing the implications of this discovery in the next version of the paper.
> >
> > References:
> >
> > [1] Haussler, David, Nick Littlestone, and Manfred K. Warmuth. "Predicting {0, 1}-functions on randomly drawn points." Information and Computation 115, no. 2 (1994): 248-292.
> >
> > [2] Farhadkhani, Sadegh, Rachid Guerraoui, and Oscar Villemaud. "An Equivalence Between Data Poisoning and Byzantine Gradient Attacks." In International Conference on Machine Learning, pp. 6284-6323. PMLR, 2022.
> >
> > [3] Rosenfeld, Elan, Ezra Winston, Pradeep Ravikumar, and Zico Kolter. "Certified robustness to label-flipping attacks via randomized smoothing." In International Conference on Machine Learning, pp. 8230-8241. PMLR, 2020.
> >
> > [4] Gao, Ji, Amin Karbasi, and Mohammad Mahmoody. "Learning and certification under instance-targeted poisoning." In Uncertainty in Artificial Intelligence, pp. 2135-2145. PMLR, 2021.
> >
> > [5] Balcan, Maria-Florina, Avrim Blum, Steve Hanneke, and Dravyansh Sharma. "Robustly-reliable learners under poisoning attacks." arXiv preprint arXiv:2203.04160 (2022).

---

### Official Review · Reviewer_qCAz · 2022-07-09

**Rating:** 7
**Confidence:** 3
**Soundness:** 3 good
**Presentation:** 3 good
**Contribution:** 3 good

**Summary:**

This work studies learnability under a targeted poisoning attack.  Specifically, they consider the best achievable error in both the realizable and agnostic settings.  Their method focuses on $\ell_0$ attacks and is fundamentally based on the ensemble-based robustness proposed by Levine and Feizi [2020].

**Questions:**

Section 2: Can you clarify what you mean when you wrote "observe that when both the learner and the output model can be randomized, and if the learner is improper, then the whole process of training and testing can be delegated to 'testing time'"?

It is not obvious to me why your method is strictly limited to only the binary classification case.  It seems it should extend to multiclass classification where the difference is w.r.t. to the two most popular classes.
* Can you comment on this multiclass case and why it seems not to apply?  Was it simply a simplification for the analysis or a technical limitation?

This paper focuses on a single target instance $x$.  Increasingly the focus is shifting to subpopulation attacks [Jagielski et al. 2022] and defenses, where the attacker targets multiple (test instances) and the defender provides certification bounds on multiple instances simultaneously.
* Have you considered how your bounds change under such scenarios?  I understand this question is not the focus of this particular work so "No" is a perfectly acceptable answer.

**Limitations:**

The authors do not include a "Limitations" section nor did I expect one for this type of theory paper.

I would have preferred some intuition in the main paper regarding the origin of the "carefully chosen" subset sizes in Figures 1 and 2.

While the writing overall was clear, the number of typos was a problem, and if this paper is accepted, it needs thorough proofreading.  Some errors are more egregious than others.
* Lines 153 & 155 "Gao et al. Gao et al." as well as "Balcan et al. Balcan et al."
* Line 284: "Besides of Prediction" -- two issues
* Line 32: "ensure the trustworthy of such"
* Line 339 & 346 regret bound
Line 39: "can can"

**Strengths And Weaknesses:**

This work follows in a long tradition of learnability under "noise" (defined broadly).   In particular, the authors address several open questions from previous work.  For example, Gao et al. prove a general bound of $o(1)$ but do not consider specific poisoning fractions, e.g., ${\eta = \frac{1}{100}}$.

Overall, I enjoyed reading this paper.  Its organizational structure was clear.  The ordering of the arguments allowed me to easily follow along without having to go back and re-read previous sections. I thought it was well done.

---

> ### Author Response · Authors · 2022-08-02
> **Answer to reviewer qCAz**
>
> We thank the reviewer for dedicating time to read and appreciate the paper and for the positive feedback!
>
> *Question #1:*
>
> Section 2: Can you clarify what you mean when you wrote "observe that when both the learner and the output model can be randomized… then the whole process of training and testing can be delegated to 'testing time'"?
>
> *Answer:*
>
> We agree that the formulation of this sentence is not clear, and we will improve it in the next version of our paper (as well as below). Thanks for pointing this out.
> Clarification: consider a learner that uses randomness both in the training process (this is training-time randomness), as well as during prediction time (this is test-time randomness).  Notice that such a learner can be simulated by a learner that only uses test-time randomness (at the cost of increased running time during prediction). This is achieved by simply delegating the randomized computation during training time to test time.  In other words, from an information theoretic perspective test-time randomness is equivalent to test+training rime randomness.
>
>
> *Question #2:*
>
> It is not obvious to me why your method is strictly limited to only the binary classification case… Can you comment on this multiclass case and why it seems not to apply? Was it simply a simplification for the analysis or a technical limitation?
>
> *Answer:*
>
> All of our results are stated in terms of the VC-dimension, which is defined for the binary case.
> To extend the results to the multiclass setting one would need to replace the VC dimension by the  Daniely-Shwartz dimension which was only recently shown to capture multiclass PAC learnability in [1].
> Extending our results to the multiclass setting in terms of the DS dimension is an interesting direction for future research, however one should note that the optimal learning rates in multiclass PAC learning (even without noise) are still not known (there is a significant gap between the upper and lower bounds in terms of the DS dimension). Thus, it seems that the first step would be to determine tight bounds in the noiseless case.
>
> *Question #3:*
>
> This paper focuses on a single target instance x. Increasingly the focus is shifting to subpopulation attacks [Jagielski et al. 2022] and defenses, where the attacker targets multiple (test instances) and the defender provides certification bounds on multiple instances simultaneously.
> Have you considered how your bounds change under such scenarios? I understand this question is not the focus of this particular work so "No" is a perfectly acceptable answer.
>
> *Answer:*
>
> As for subpopulation attacks, indeed we did not consider this extension in our work and we agree that it is a great direction for future research.
>
> *Limitations:*
>
> The authors do not include a "Limitations" section nor did I expect one for this type of theory paper. I would have preferred some intuition in the main paper regarding the origin of the "carefully chosen" subset sizes in Figures 1 and 2. While the writing overall was clear, the number of typos was a problem, and if this paper is accepted, it needs thorough proofreading.
>
> *Answer:*
>
> We completely agree and thank the reviewer for pointing these out. We will proofread the paper, fix the typos, and attempt to add intuition regarding the subsets sizes.
>
> References:
>
> [1] Brukhim, N., Carmon, D., Dinur, I., Moran, S., & Yehudayoff, A. (2022). A Characterization of Multiclass Learnability. arXiv preprint arXiv:2203.01550.

---

> > ### Comment · Reviewer_qCAz · 2022-08-02
> > **Rebuttal Acknowledgment**
> >
> > Thank you for your detailed response. I have read it along with your responses to the other reviewers. I will make a final decision on the score after reviewing any further discussion with the other reviewers.  At this point, I have no additional questions.

---

> > > ### Comment · Reviewer_qCAz · 2022-08-08
> > > **Updated Score**
> > >
> > > There was no additional discussion during the interactive period. With the feedback window soon to close, I have increased my score, and I believe this paper would be a valuable contribution to NeurIPS.

---

### Official Review · Reviewer_nSt5 · 2022-07-09

**Rating:** 6
**Confidence:** 3
**Soundness:** 3 good
**Presentation:** 3 good
**Contribution:** 3 good

**Summary:**

This work studies the optimal rate of learning for binary settings under a fraction of $\eta$ poisoning data under targeted data poisoning attack. Under realizable setting, the error rate is proportional to $\eta$ and the VC dimension of the hypothesis set, which can be attained by a deterministic learner. Under agnostic setting, for any deterministic learner, a lower bound is present to show an inevitable multiplicative deterioration in the regret.

**Questions:**

1. I’m wondering whether the bound would change if we switch the poisoning attack to non-targeted poisoning attacks. Also, whether there’s any concrete algorithm for the targeted poisoning attack.

2. For the agnostic cases lower bound, I’m wondering whether considering certain noise settings (i.e. Massart noise, Tsybakov noise) will eliminate the additional multiplicator on OPT.

3. For sec 3.2, I’m wondering whether there’s a lower bound when considering proper learner.

4. The author has a discussion on the potential role of randomness on page 5, but I think it should be more clear if first put more work into describing deterministic vs randomness in terms of learning rules and models by either giving examples or giving formal definitions. In the middle of the discussion, the author then mixed randomness and improper learning together. I understand the author may express the idea of improper learners like majority vote involves randomness in the model or learning rule, but it’s unclear currently from the text what’s the take-home message.  Moreover, if the learning rule introduces any randomness,  I’m wondering whether there’s any literature studying the scenario where the attacker can also get access to the random seed.

5. For the discussion of prediction stability, I think the author is trying to make a difference between robustness and astuteness, which I suppose has been clarified in [1] before. Though here the context is under data poisoning attack, since the idea is the same, it might be good to make a comparison.

6. In section 4, it'd be helpful to give a formal definition of semi-agnostic learning.

7. In figure 2 PSPV, what’s the definition of $X_{\{h_1,\ldots,h_t\},2k_p}$. Moreover, does PSPV only limited to linear classifiers or there are no constraints on the hypothesis set. Since the only difference between SPV and PSPV is the latter output $h\in\mathcal{H}$, yet both of $h$ are computed via $Maj$, then how can you guarantee the hypothesis computed in PSPV belongs to $\mathcal{H}$?

**Limitations:**

Yes.

**Strengths And Weaknesses:**

Strengths:
1. This work shows that under the realizable case the adversarial risk bound is tight in terms of corruption fraction and VC dimension of the hypothesis set.
2. This work provides a lower bound on the adversarial error for agnostic setting.


Weaknesses:
1. I find the poisoning scenario that is described a bit confusing. Notice that targeted poisoning can also be regarded as selecting a targeted label $y_t$, and adding poisoning data to make the test sample misclassify as the targeted label $y_t$. But here to my understanding, the targeted poisoning is the adversary can see the test sample ahead of time so that the adversary can add poisoning data to make the test sample misclassify.

---

> ### Author Response · Authors · 2022-08-02
> **Answer to reviewer nSt5: main body**
>
> We thank the reviewer for dedicating time to assess our work and for providing constructive suggestions.
>
> *Weakness #1:*
>
>  “I find the poisoning scenario that is described a bit confusing. Notice that targeted poisoning can also be regarded as selecting a targeted label $y_t$, and adding poisoning data to make the test sample misclassify as the targeted label $y_t$….”
>
> *Response:*
>
> The form of targeted poisoning used in our work is known as instance-targeted-poisoning (see e.g. [3,4] below). It is probably confusing that we used the shorter term “targeted poisoning” instead of “instance-targeted poisoning”. We will fix this in the next version of the paper and add a discussion surveying the different forms of targeted poisoning.
>
> *Question #1:*
>
> “I’m wondering whether the bound would change if we switch the poisoning attack to non-targeted poisoning attacks. Also, whether there’s any concrete algorithm for the targeted poisoning attack.”
>
> *Answer:*
>
> Yes, the bound will change to ~$\eta$ instead of ~$d \eta$ for targeted poisoning attacks (see [1]). Our attack exploits a specific "hard" distribution, for which it attains the maximal risk (up to a constant multiplicative factor). We did not phrase the attack as an algorithm, mostly because it attains the maximal risk for a specific hard distribution. Coming up with an algorithm providing an optimal attack for a given distribution is an interesting problem, but is beyond the scope of our work.
>
> *Question #2:*
>
> For the agnostic cases lower bound, I’m wondering whether considering certain noise settings (i.e. Massart noise, Tsybakov noise) will eliminate the additional multiplicator on OPT.
>
> *Answer:*
>
> This is an interesting direction, but we did not consider it in this work.
> (Notice that it is even open whether a randomized learning algorithm might eliminate the multiplicative regret guarantee.)
> We refer the reviewer to our general response for further clarification regarding our focus in the agnostic setting.
>
> *Question #3:*
>
> For sec 3.2, I’m wondering whether there’s a lower bound when considering proper learner.
>
> *Answer:*
>
> The best lower bound we found for proper learning is the same as for improper learning. Shrinking the gap between proper learning and improper learning (even just for linear classifiers) remains an interesting open question.
>
> *Question #4:*
>
> The author has a discussion on the potential role of randomness on page 5, but I think it should be more clear if first put more work into describing deterministic vs randomness in terms of learning rules and models… In the middle of the discussion, the author then mixed randomness and improper learning together… it’s unclear currently from the text what’s the take-home message…
>
> *Answer:*
>
> Thanks for pointing this out. We see now that this discussion should be clarified, and we will do so in the next revision of the paper.  The take-home message is as follows. For randomized algorithms one can consider two types of adversaries: a strong one which sees the internal randomness of the learner and a weak one which does not. The point we were trying to make is that our result is as strong as it can be in the sense that our upper bound is achieved by an algorithm which is robust against the *strong* adversary (which has access to the learner’s randomness), whereas our lower bound is achieved by a weak adversary (who does not have access to the learner’s randomness).
>
> *Question #5:*
>
> For the discussion of prediction stability, I think the author is trying to make a difference between robustness and astuteness, which I suppose has been clarified in [1] before… it might be good to make a comparison.
>
> *Answer:*
> Please provide us with more details so we can best address this question. Specifically, we are not sure which paper the reference “[1]” was meant to cite.
>
>
> *Question #6:*
>
> In section 4, it'd be helpful to give a formal definition of semi-agnostic learning.
>
> *Answer:*
> Thanks, we agree. We will add it to the next version of the paper.
>
> *Question #7:*
>
> In figure 2 PSPV, what’s the definition of $X_{\{h1,…,ht\},2kp}$?…. Since the only difference between SPV and PSPV is the latter output $h \in \mathcal{H}$, yet both of $h$ are computed via $Maj$, then how can you guarantee the hypothesis computed in PSPV belongs to $\mathcal{H}$?
>
> *Answer:*
>
> The definition of the set $X_{\{h1,…,ht\},2kp}$ appears in Definition 3.7 (Projection number). It is guaranteed that $h$ belongs to the hypothesis set, directly by the definition of projection number. So, PSPV will work for any hypothesis set having a finite projection number. The interesting part is that there are known and important hypothesis classes with a finite projection number. As an example, we give the class of halfspaces over $\mathbb{R}^d$ which has a projection number d+1. This is a result by [2], appearing as Theorem C.2 in the appendix.

---

> > ### Author Response · Authors · 2022-08-02
> > **Answer to reviewer nSt5: references**
> >
> > References:
> >
> > [1] Bshouty, Nader H., Nadav Eiron, and Eyal Kushilevitz. "PAC learning with nasty noise." Theoretical Computer Science 288, no. 2 (2002): 255-275.
> >
> > [2] Bousquet, Olivier, Steve Hanneke, Shay Moran, and Nikita Zhivotovskiy. "Proper learning, Helly number, and an optimal SVM bound." In Conference on Learning Theory, pp. 582-609. PMLR, 2020.
> >
> > [3] Gao, Ji, Amin Karbasi, and Mohammad Mahmoody. "Learning and certification under instance-targeted poisoning." In Uncertainty in Artificial Intelligence, pp. 2135-2145. PMLR, 2021.
> >
> > [4] Balcan, Maria-Florina, Avrim Blum, Steve Hanneke, and Dravyansh Sharma. "Robustly-reliable learners under poisoning attacks." arXiv preprint arXiv:2203.04160 (2022).

---

### Official Review · Reviewer_t9Qw · 2022-07-11

**Rating:** 6
**Confidence:** 3
**Soundness:** 4 excellent
**Presentation:** 4 excellent
**Contribution:** 4 excellent

**Summary:**

This paper studied developing robust algorithms in presence of an attacker who can corrupt a portion of the training set in the classical version space learning scenario. In the realizable scenario, the paper provided impossibility result that shows if the attacker can perturb eta fraction of training samples, then the test accuracy cannot be lower than eta*d with high probability, where d is the VC dimension of the hypothesis class. To complement this result, the paper then proposed robust PAC learning algorithms that provably achieve this upper bound, which perfectly closes the robust learning problem in the realizable case. In the agnostic scenario, the paper showed that no algorithms can achieve 2*OPT test accuracy, where OPT is the optimal error rate without poisoning. Then the paper proposed a robust algorithm that achieves O(OPT)+delta error, where delta depends on the VC dimension.

**Questions:**

(1). Whether the threat model considered in this paper is targeted or untargeted poisoning attack?

(2). How tight is the error upper bound in the agnostic scenario?

**Limitations:**

Yes

**Strengths And Weaknesses:**

Strengths:

(1). The paper provided a very comprehensive study of PAC learning under data poisoning attacks. When the attacker can corrupt eta fraction of the training set, the paper showed impossibility result and proposed robust PAC learning algorithms that achieve the error lower bound. This result is fundamental as it extends PAC analysis from traditional machine learning to the robust machine learning domain.

(2). The theoretical results developed in this paper is important. The proof techniques are novel and can be used to study related topics in future research. The topic is extremely fundamental in the machine learning community, which help researchers builds deep understanding of data poisoning attacks.

(3). The paper is well-written and the structure is articulated in a very clear way. The annotation goes from realizable scenario to the agnostic scenario, which makes the paper easier to read and understand.

Weaknesses:

(1). I think the threat model studied in this paper is "untargeted data poisoning attack" instead of "targeted poisoning attack". In typical targeted poisoning attack, the goal of the attacker is to change the prediction on a test example to a pre-specified target label. However this paper considered poisoning attacks that aim at reducing the test-time accuracy overall, which is often consider as untargeted poisoning attacks. I suggest the authors check if the terminology is aligned with the SOTA literature.

(2). In the agnostic scenario, while the paper proved that no learners can achieve 2OPT error rate under data poisoning attacks. It's not very clear how the upper bound O(OPT)+delta compares to the lower bound. That said, I am not sure if the upper bound achieved by the proposed robust PAC learner is tight or not. I hope that the authors could elaborate more on this point.

---

> ### Author Response · Authors · 2022-08-02
> **Answer to reviewer t9Qw**
>
> We thank the reviewer for the positive review and for commenting that: “The theoretical results developed in this paper is important. The proof techniques are novel and can be used to study related topics in future research. The topic is extremely fundamental in the machine learning community, which help researchers builds deep understanding of data poisoning attacks.” Given the positive content of the review we respectfully ask the reviewer to consider raising the score.
>
> The reviewer raise two specific questions:
>
> *(1). Whether the threat model considered in this paper is targeted or untargeted poisoning attack?*
>
> *(2). How tight is the error upper bound in the agnostic scenario?*
>
> *Answer to question #1:*
>
> The reviewer is completely right that there are multiple ways the term “targeted” has been defined in the literature. The form the reviewer refers to is called “label-targeted”, whereas what we mean is “instance-targeted” (see, e.g. [1,2,3]). Another form which is called “model-targeted” is mentioned in the review by dWRe.
>
> We see now that it is confusing that we used the shorter term “targeted poisoning” instead of “instance-targeted poisoning”. We will fix this in the next version of the paper and add a discussion clarifying the other different forms of targeted-poisioning. Thanks!
>
> On the other hand, the term “untargeted attacks” refers to an adversary whose goal is to increase the overall risk of the algorithm. (In particular, the adversary does not have a particular target label/instance in mind.)
> This model was studied already a few decades ago (see [4,5] for example).
>
> *Answer to question #2:*
>
> The constant C which follows from our analysis of the upper bound is  approximately C = ~90. One can decrease it by complicating the proof a bit, but not significantly. Shrinking the gap between 2 and C is one of the main open questions posed in the paper.
>
> Nevertheless we would like to stress that this question is somewhat beyond the main focus of the paper.  We refer the reader to our general response for further clarification on this.
>
> References:
>
> [1] Rosenfeld, Elan, Ezra Winston, Pradeep Ravikumar, and Zico Kolter. "Certified robustness to label-flipping attacks via randomized smoothing." In International Conference on Machine Learning, pp. 8230-8241. PMLR, 2020.
>
> [2] Gao, Ji, Amin Karbasi, and Mohammad Mahmoody. "Learning and certification under instance-targeted poisoning." In Uncertainty in Artificial Intelligence, pp. 2135-2145. PMLR, 2021.
>
> [3] Balcan, Maria-Florina, Avrim Blum, Steve Hanneke, and Dravyansh Sharma. "Robustly-reliable learners under poisoning attacks." arXiv preprint arXiv:2203.04160 (2022).
>
> [4] Valiant, Leslie G. "Learning Disjunction of Conjunctions." In IJCAI, pp. 560-566. 1985.
>
> [5] Bshouty, Nader H., Nadav Eiron, and Eyal Kushilevitz. "PAC learning with nasty noise." Theoretical Computer Science 288, no. 2 (2002): 255-275.

---

### Author Response · Authors · 2022-08-02
**General response to all reviewers**

We thank the reviewers for taking the time to carefully read our work, and for their thoughtful comments and suggestions to improve it.

The reviewers seem to agree that the main result in this paper is fundamental (e.g. that it extends the classical characterization of optimal PAC learning rates to robust learning rates under instance-targeted-poisoning). Apart from relatively minor comments regarding presentation and terminology (which we address in dedicated replies), the main issue the reviewers raise is the gap between our upper and lower bounds in the agnostic case.

Therefore, we would like to stress that our main aim in this work is addressing the basic, categorical question of  which classes can be learned under targeted attacks and what are the optimal robust learning rates. As the reviewers noted, we completely resolved this question in the realizable setting.

In the agnostic setting our impossibility result implies that for deterministic algorithms, agnostic learning is in fact *impossible*. Indeed, the standard definition of agnostic learning requires a learning rule which achieves a negligible additive error compared to the best hypothesis in class. In contrast, we show that unless the class is trivial (i.e. contains at most one function), there exists an attack which increases the error by a *multiplicative* factor of at least 2.

Thus, our overarching question is also answered in the agnostic case, at least for deterministic learning rules. We then continue to explore the natural (but non-standard) variant of agnostic learnability, where one allows multiplicative regret. Perhaps surprisingly, we exhibit a deterministic learner achieving this.

Another contribution of this work is that it encourages further research by leaving several fundamental open questions; e.g. (i) Is agnostic learning possible with randomized learners?, (ii) what is the best factor possible in the multiplicative regret?

We hope that this helps clarify the scope of this paper and puts our results regarding the agnostic setting in better context. We believe the reviewers agree that the overarching question guiding this work is fundamental, and see that we resolved it in the realizable case and in the deterministic agnostic case. A complete resolution of the agnostic case is left for future work which we hope our work will promote and inspire. We respectfully ask the reviewers to revise their scores, especially since they seem to agree on the fundamental nature of our results.

---

### Meta-Review · Area_Chair_mpfK · 2022-08-27

**Recommendation:** Accept
**Confidence:** Certain

**Metareview:**

All reviewers agree that this paper comprehensively studies a fundamental question of PAC learning under instance-targeted poisoning, including the study of realizable, agnostic, deterministic learning settings; overall this paper makes a nice contribution to the field of robust machine learning.

The authors are strongly encouraged to incorporate the comments by the reviewers, including revising on motivating examples, terminologies, etc.

**Award:**

No

---

### Decision · Program_Chairs · 2022-09-14

Accept